



# The impact of assimilating Aeolus wind data on regional Aeolian dust model simulations using WRF-Chem

Pantelis Kiriakidis[1], Antonis Gkikas[2], George Papangelis[2], Theodoros Christoudias[1], Jonilda Kushta[1], Emmanouil Proestakis[2], Anna Kampouri[2], Eleni Marinou[2], Eleni Drakaki[2], Angela Benedetti[3], Michael Rennie[3], Christian Retscher[5], Anne Grete Straume[4], Alexandru Dandocsi[5], Jean Sciare[1], and Vasilis Amiridis[2]

[1]Climate and Atmosphere Research Center, The Cyprus Institute, 2121 Nicosia, Cyprus
[2]National Observatory of Athens, Nymphs Hill 118 10, Athens, Greece
[3]European Centre for Medium Range Weather Forecasts, RG2 9AX, Reading, United Kingdom
[4]European Space Agency, 2201 AZ, Noordwijk, Netherlands
[5]European Space Agency, 00044, Frascati, Italy

**Correspondence:** Theodoros Christoudias (t.christoudias@cyi.ac.cy)

**Abstract.**

Land-atmosphere interactions govern the process of dust emission and transport. An accurate depiction of these physical processes within numerical weather prediction (NWP) models allows for better estimating the spatial and temporal distribution of the dust burden and the characterisation of source and recipient areas. In the presented study, the ECMWF-IFS (European Centre for Medium-Range Weather Forecast - Integrated Forecasting System) outputs are used to simulate two-month long periods in the spring and autumn of 2020, focusing on a case study in October. The ECMWF-IFS outputs are produced with and without assimilation of Aeolus quality-assured Rayleigh-clear and Mie-cloudy Horizontal Line of Sight (HLOS) wind profiles. The experiments have been performed over the broader Eastern Mediterranean and Middle East (EMME) region that is frequently subjected to dust transport, as it encompasses some of the most active erodible dust sources. Aerosol and dust-related model outputs (extinction coefficient, optical depth and concentrations) are qualitatively and quantitatively evaluated against ground- and satellite-based observations. Ground-based columnar and vertically resolved aerosol optical properties are acquired through AERONET sun photometers and Polly[XT] lidar, while near-surface concentrations are taken from EMEP. Satellite-derived vertical dust and columnar aerosol optical properties are acquired through LIVAS and MIDAS, respectively.

Overall, in cases of either high or low aerosol loadings, the model predictive skill is improved when WRF simulations are initialised with IFS meteorological fields in which Aeolus wind profiles have been assimilated. The improvement varies in space and time, with the most significant impact observed for the autumn months in the study region. Comparison with observation datasets saw a remarkable improvement in columnar aerosol optical depths, vertically resolved dust mass concentrations and near-surface particulate concentrations in the assimilated run against the control run. Reductions of model biases, either positive or negative, and an increase in the correlation between simulated and observed values were achieved.



## 1 Introduction

The Levantine basin and Eastern Medirraenean are frequent recipients of dust transported from North Africa and the Middle East (Gkikas, 2013, 2016), receiving an estimated annual influx of 40-150 g/m$^2$ (Ben-Asher, 2019). Even though dust deposition can enhance the growth of terrestrial and oceanic ecosystems, high dust loads can have severe implications for human health, increasing the probability of a population developing respiratory and cardiovascular disease (Kanatani, 2010; Xu, 2019). Additionally, dust storms affect the built environment by degrading the life of electrical equipment, impacting several modes of transport, decreasing the efficiency of solar-harnessing technologies and severely damaging crop output (Hachicha, 2019; Middleton, 2017; Weinzierl, 2012; Stefanski, 2009).

Dust sources are characterised as areas with high availability of alluvium silts and high wind speeds. Some of the most critical Sahelian dust sources include the Bodélé and the Mauritania-Mali locality, both characterised as large basins of internal drainage, with the former being responsible for the production of 6-18% of global dust emissions (Todd, 2007; Engelstaedter, 2007). A high moisture content, the presence of physical and biological crusts and high surface roughness, dependent on vegetation cover, reduce the exposure of erodible surfaces and increase cohesion (Gillette, 1989; Ravi, 2011). The combination of these factors determines the threshold velocity, which, if exceeded, causes dust to become uplifted. Once suspended, dust from North Africa can reach as far as the Caribbean (Prospero, 1999, 2014) depositing 261±48 Tg along its path (Ridley, 2012). Approximately 30-50% of the dust emitted from the Sahel is transported through the North Atlantic trajectory (Prospero, 1996) during the summer months. Meanwhile, northward branches carrying dust in West Europe peak during late spring and summer, acting in parallel with an eastward trajectory transporting dust towards the EMME region (Middleton, 2001). In recent years, an increasing contribution of dust loads in the EMME region from Middle Eastern sources has been highlighted (De-Châtel, 2014; Pozzer, 2015; Notaro, 2015; Kelley, 2015; Logothetis, 2021) and has been attributed to changing climatic conditions. Specifically, the El-Nino Southern Oscillation and the Pacific Decadal Oscillation teleconnections (Kelley, 2015; Pozzer, 2015) have been linked to a regime shift in dust activity of the Fertile Crescent area in Mesopotamia (Notaro, 2015). Prolonged periods of droughts subsequently decreased soil moisture, where in the work of Klingmüller (2016), an R$^2$ of 0.82-0.89 was established between decreased soil moisture and the observed positive trend in dust activity for Saudi Arabia and Iraq. Additionally, Gkikas (2022) identified seasonal variability in the dust optical depth over Mesopotamia, with peaks occurring during the spring and summer dry periods. Dust storm activity in Mesopotamia and Saudi Arabia is most frequent during spring when eastward moving, steep pressure gradients cross through the Mediterranean into Turkey and Iraq (Middleton, 1986). Alongside convective cold pools and nocturnal low-level jets, these mechanisms are responsible for 80% of dust transport during late spring and early summer (Heinold, 2013; Chedin, 2018). Additionally, the Mediterranean region hosts its highest anticyclonic activity during winter and the lowest during autumn (Godev, 1971). These extratropical migratory anticyclones have been characterised to be of synoptic and sub-synoptic scale, with slight deviations from their initial position towards the north-east (Hatzaki, 2014; Ioannidou, 2008). Throughout all seasons, the western North African region is typically the source of the anticyclones, while areas within the East Mediterranean, such as Greece, have been identified as sinks (Lolis, 2014; Trigo, 2004).



The continuous improvements in the computational capability of NWP models and the growth of available high-resolution observations have significantly improved the accuracy of dust episode forecasting. Numerous sources of uncertainty still prevail, specifically the description of the dust source function and wind fields within an NWP model. Previous work improving the former source of uncertainty has been demonstrated in global models through the works of Ginoux (2001), Zender (2003), Schepanski (2009) and Nabavi (2017), meanwhile Kok (2014) and Wu (2016), have achieved similar improvements by including a threshold velocity parameterisation. However, significant variability in total dust emissions within global models persists, with emissions ranging from 500 to 6000 Tg/yr (Ginoux, 2001; Huneeus, 2011; Prospero, 2010), attributed to differing model parameterisations and configurations used (Uno, 2006; Huneeus, 2011). In the studies of Alonso-Pérez (2012) and Cavazos-Guerra (2012), a Lagrangian-Eulerian model and a regional WRF-Chem (Weather Research and Forecasting model with Chemistry) model, respectively, were assessed in terms of dust emissions constrained to West Sahara sources. In the former study, the period of 1998-2003 was investigated, and in the latter, two months in 2006 were simulated. Both study periods pre-date the recent negative dust trend observed for dust sources of Saharan origin (Mehta, 2016; Chedin, 2018; Shaheen, 2021). Tegen (2013) using the COSMOS (Consortium for Small scale Modelling) meteorological model for the years 2007 and 2008 for the Sahara region, identified challenges in the model ability to accurately depict inter-annual variability due to gaps in understanding controls of the atmospheric dust load. Nabavi (2016) identified existing challenges of dust source functions within the WRF-Chem model unable to accurately depict dust sources in Mesopotamia, an area of increasing importance in dust studies. Following, Nabavi (2017) tested a new function termed WASF (West Africa source function) within the WRF-Chem model, aiming to improve multi-year analysis for the summers of 2008–2012. Even though the incorporation of the WASF significantly improved the Spearman correlation, the accuracy of forecasts dropped with increasing distance from the source. This signified deficiencies in model transport and deposition mechanisms (Nabavi, 2017).

The WRF-Chem model allows the implementation of various parameterisation schemes for dust modelling, with numerous studies enhancing the model to be particularly suitable for the EMME region. Tsarpalis (2018) assessed the impact of a deposition scheme using a dust outbreak in June 2014 affecting the Eastern Mediterranean. It was concluded that reductions in overestimations of dust residing at heights greater than 1 km could be achieved. However, underestimations increased at the surface level. Rizza (2018) simulated a dust outbreak in March 2016 at the Central Mediterranean and inter-compared three soil property models. The NoahMP model was performing the best, but in all three runs, the dust peak was time-shifted relative to the observed peak. Flaounas (2017) studied months of peak activity for the year 2011, with three dust emission parameterisation schemes, highlighting the importance of long-term simulations for sensitivity testing. Compared to the targeted sensitivity testing of the WRF-Chem model performed for the Central and East Asia regions, a gap exists in inter-comparison studies over the EMME region, particularly in assessing individual dust simulation components (Darmenova, 2009; Kang, 2011; Su, 2015; Yuan, 2019; Zeng, 2020; Zhao, 2020).

The model predictive ability also benefits from an observational coverage network able to feed the model with observations. Even though there is broad coverage of ground-based stations along the European-Mediterranean border, this does not stand true for the North-African border and especially for the turmoiled Middle East region. The shortcoming of ground-based observational data can be combated through the use of ever-increasing satellite products, one being the recently launched Aeolus



Doppler wind lidar. The European Space Agency (ESA), in August 2018, launched the Aeolus satellite carrying ALADIN,
the first-ever space-based lidar. ALADIN, via the HSRL technique (Shipley, 1983), acquires HLOS wind profiles up to 30 km
all over the globe. Thus, advancing the current poor observational capabilities, particularly in the open seas of the Southern
Hemisphere and the vast desert areas. The first assessment studies (Baars, 2020; Lux, 2020; Witschas, 2020) of Aeolus wind
products during the satellite commission phase (autumn 2018) revealed the capability of Aeolus to derive high-quality wind
profiles. The main scientific goals of the Aeolus satellite mission are to advance NWP and upgrade the current level of knowl-
edge on atmospheric dynamics and their associated impacts on climate (Stoffelen, 2005; Isaksen, 2019; Rennie, 2019). The
positive impact of Aeolus wind data assimilation on NWP has been demonstrated by the ECMWF starting the operational as-
similation of Aeolus L2B wind data on January 2020 (Baars, 2020). Such activities have also been adopted by other European
weather forecast centres (DWD, Météo-France and UK MetOffice). Rennie (2021), demonstrated the beneficial impact of Ae-
olus wind assimilation on the short- and medium-term forecasts in the S. Hemisphere, in polar regions as well as in latitudinal
bands where the well-developed "dust belt" stretches (Prospero, 2002). Since winds trigger dust mobilisation and drive the
advection patterns of dust plumes, a subsequent positive impact of Aeolus wind assimilation on numerical dust simulations is
anticipated. This improvement constitutes the overarching objective of the NEWTON (ImproviNg dust monitoring and forE-
casting through Aeolus Wind daTa assimilation; https://newton.space.noa.gr/) project funded by the ESA in the framework of
the Aeolus+ Innovation call. In this study, the ECMWF-IFS assimilated Aeolus wind fields provided by ESA are implemented
in the WRF-Chem to study the effect on the simulated dust. The model is initialised with two sets of IFS outputs that differ
only in consideration of Aeolus wind profiles in the respective assimilation scheme. Four WRF-Chem runs were produced for
April - May and September – November 2020, capturing the dust seasons of the region.

This article is structured as follows, in Sects. 2.1 and 2.5, the regional WRF-Chem model setup is described, alongside the
observation datasets and the collocation methodologies undertaken, respectively. Following, Sect. 3.1 describes the meteoro-
logical conditions simulated using the ECMWF-IFS products with and without Aeolus. Finally, in Sects. 3.2 and 3.3 model
outputs are compared against surface measurements provided by AERONET, EMEP and Polly[XT] and satellite-based MIDAS
and LIVAS, leading to conclusions in Sect. 4.

## 2   Data and Methodology

In the following section, the WRF-Chem regional model configuration and the incorporation of the Aeolus assimilated wind
fields within the ECMWF-IFS datasets are discussed in 2.1 and 2.2 respectively. Ground- and satellite-based observation
datasets used to evaluate the model are discussed in section 2.5, alongside the methods employed for their spatio-temporal
collocation to simulated outputs.

### 2.1   WRF-Chem Model Setup

The WRF-Chem version 3.9.1.1 was used along with the WRF Pre-Processing System version 4.2 to perform meteorological
and air quality simulations adjusted for the study region.



The RACM mechanism was applied to simulate gas-phase chemistry. RACM is based on the Regional Acid Deposition Model, version 2 mechanism and has been shown by Georgiou (2018) that it produces the lowest Mean Bias for hourly concentrations of fine particles over the region of interest compared to other gas-phase chemistry mechanisms. The Modal Aerosol Dynamics Model for Europe (MADE) and the secondary organic aerosol (SOA) parameterisation based on the volatility basis
set by Ahmadov (2012) were employed to simulate aerosol inorganic species and SOA, respectively. Anthropogenic emissions were based on the EDGAR-HTAP (Emission Database for Global Atmospheric Research for Hemispheric Transport of Air Pollution) Version 5 emission inventory compiled by the European Commission, Joint Research Centre/Netherlands Environmental Assessment Agency (Janssens-Maenhout, 2012). The Model of Emissions of Gases and Aerosols from Nature version 2.1 (MEGAN2.1) by Guenther (2012) was employed to generate biogenic emissions based on weather and land use data.

Natural emissions were calculated on-line by the WRF-Chem model, driven by IFS Aeolus-assimilated data. The Georgia Tech/Goddard Global Ozone Chemistry Aerosol Radiation and Transport (GOCART) model (Ginoux, 2001), coupled with the MADE/SORGAM aerosol mechanism within the framework of WRF-Chem, was used to simulate dust emission. The dust emission flux in the GOCART model is scaled by an empirical proportionality constant C. The value of C, estimated by Ginoux (2001), was initially based on US regional data. Zhao (2010) evaluated the performance of the WRF-Chem model
for different values of C. They found that for C = 0.4 $\mu$gs$^2$/m$^5$, the WRF-Chem simulated mean aerosol optical depth (AOD) was consistent with the AERONET measurements at two sites over the Sahel region and aerosol size treatments over North Africa (Zhao, 2010). Several sensitivity tests performed over Cyprus and the East Mediterranean led to the best performing value of 0.36 $\mu$gs$^2$/m$^5$ (Georgiou, 2018), based on the study of modelling sensitivities to dust emissions. Therefore, as this is the most prevalent source of dust emissions in the EMME region, in the following simulations, a value of C equal to 0.36
$\mu$gs$^2$/m$^5$ was used. The radiation scheme RRTMG has been used due to the incorporation of a two-stream radiative transfer solver (Oreopoulos, 1999) that enables a more accurate calculation of extinction from aerosols in the presence of multiple scattering (Iacono, 2008). A summary of the model configuration options is presented in Table A1 of the Supplement.

## 2.2  Aeolus assimilation

The WRF numerical experiments are performed using the simulated meteorological fields from the IFS as initial and boundary
conditions. The IFS refers to a global numerical weather prediction system of the ECMWF. Based on the defined workflow, the control and assimilated runs produce two sets of IFS outputs. In both simulations, various observations are assimilated to reproduce the optimum state of the atmosphere (i.e., analysis). For the analyses at 00 UTC and 12 UTC, all the quality-assured observations falling within the time windows spanning from 21 UTC (the day before) to 09 UTC and from 09 UTC to 21 UTC, respectively, are assimilated. The assimilation is done with the 4D-Var technique via a process called LWDA (Long Window
Data Analysis; formerly known as "delayed cut-off", Haseler (2004)), which allows the maximum use of available data. The production of the analysis increments considers all observations falling within the assimilation time window. Following, short-term forecasts are initialised at the start of the time window (either 21 UTC or 09 UTC), which correspond to the analysis fields at 00 UTC and 12 UTC and to intermediate nominal model times (i.e., 06 UTC and 18 UTC).



In contrast to the control IFS run, the assimilated run includes the Aeolus L2B (2B10 baseline) Rayleigh-clear and Mie-
cloudy HLOS wind profiles (Baars, 2020; Rennie, 2019). The quality screening of Aeolus HLOS retrievals is identical to
those applied in the OSEs (Observing System Experiments) in Rennie (2021), who provided the configuration of the IFS runs
(see the second row in Table 1 in Rennie (2021)), utilised in the current study. Finally, the two sets of IFS numerical outputs
are produced on a 3-hourly basis and serve as initial and boundary conditions in the WRF-Chem model simulations. The
simulations are contrasted to investigate the modifications in the meteorological patterns and the subsequent variations in the
simulated dust fields.

## 2.3 Experiments set-up

The WRF-Chem model was used to simulate the periods 2020/03/04 - 2020/05/31 and 2020/09/01 - 2020/11/04 using boundary
conditions from IFS with (assimilated) and without (control) Aeolus assimilated data. The periods under investigation coincide
with the dust storm high-activity phase of the East Mediterranean (Engelstaedter, 2006; Miller, 2008; Tyrlis, 2014). The domain
area has been configured to a horizontal grid resolution of 20 x 20 km, extending from 5° to 45° in latitude and -20° to 62.5°
in longitude, spanning over the three primary dust sources affecting the region (domain area visualised in Fig. A1 of the
Supplement). Including the Saharan, Syrian and Arabian deserts in the model domain allows to not require the dust component
from the boundary conditions and helps avoid biases in dust concentrations from global models. The WRF-Chem model uses
a terrain-following hydrostatic-pressure vertical coordinate system. The model configuration uses 30 layers, from the surface
up to 50 hPa, with an average height of 70 m for the first layer. Additionally, the WRF-Chem model has Four-Dimensional
Data Assimilation (FDDA) capabilities, described in Deng (2012). By employing the FDDA, the continuous assimilation of
meteorological observations in the model is enabled. In turn, this can improve the lateral boundary conditions and the initial
state of the meteorological conditions, thereby improving the short-term forecast (Reen, 2010; Deng, 2012; Rogers, 2013). In
the study of Deng (2007), it was shown that by nudging above and within the Planetary Boundary Layer, the accuracy of the
meteorological variables simulated within the WRF-Chem model is improved and has since been used in other dust-related
studies (e.g. Kumar (2014)). Following this reasoning, the horizontal wind components, temperature and moisture were nudged
for all the model vertical layers with a nudging coefficient of $3 \times 10^{-4}$ s$^{-1}$.

To evaluate the performance of WRF-Chem with the introduction of the assimilated ECMWF-IFS dataset, we employ the
total-column atmospheric extinction coefficient at the wavelength of 550 nm (EC55). EC55 can be used as a proxy for dust
optical depth during dust outbreaks as calculated through the radiation scheme RRTMG.

## 2.4 FLEXPART

For the characterisation of the origins of air masses, the Lagrangian particle dispersion model FLEXPART (Stohl, 2005;
Brioude, 2013; Pisso, 2019) was run in a backward mode for the period 14th to 25th October 2020. The backward FLEXPART-
WRF runs were performed by releasing 10,000 tracer particles at heights 0.5, 1, 2, 3, 4, 5, 7 and 10 km over the Agia Marina,
Cyprus station. The FLEXPART simulations were driven by hourly meteorological fields from the WRF-Chem model initiated
with control and assimilated datasets.



## 2.5 Observation Datasets

To evaluate the performance of the WRF-Chem model, ground-based; AERONET AOD, EMEP coarse particulate matter (PM10) and Polly[XT] vertical dust concentrations were used. These were accompanied by space-borne horizontal and verti-
cal dust products from MIDAS and LIVAS, respectively. Spatial and temporal collocations were applied to enable a direct comparison between simulated and observed variables, which are discussed in the following sub-sections.

### 2.5.1 AERONET

AERONET (AEronet RObotic NETwork) is a global network of about 1000 ground-based monitoring instruments distributed globally and maintained by NASA (Holben, 1998). The sunphotometer instruments measure the spectral AOD (a unitless
measure of aerosol load throughout the total atmospheric column), aerosol size parameters (e.g., Ångström exponent) and several other optical and microphysical properties (Dubovik, 2000, 2006). Due to the limited availability of Level 2, cloud-screened and quality-assured data, Level 1.5 (only cloud-screened) AOD measurements (Version 3, (Giles, 2019; Sinyuk, 2020)) are taken from 56 stations located within the WRF-Chem constructed domain.

The WRF-Chem model does not directly output the AOD and thus has to be calculated using Equations 1 and 2. Where H
refers to height (in km), $\Delta H$ to change in height, EC55 to the total atmospheric extinction coefficient at 550 nm (in km$^{-1}$), PH to perturbation geopotential (in m$^2$/s$^2$), PHB to base-state geopotential (in m$^2$/s$^2$), V to the vertical layer, t to the time interval, lat to latitude and lon to longitude coordinates. The resulting AOD refers to the wavelength at 550 nm. AERONET AOD retrievals at this wavelength are not readily available. To make results comparable, AERONET AOD$_{870}$ has been converted to AOD$_{550}$ using the Ångström Exponent equation and the Ångström exponent of wavelength 440-870 nm (Angström,
1929). A statistical comparison of AOD$_{550}$ produced from various wavelengths was carried out, revealing negligible statistical differences regardless of the wavelength used in the conversion, also highlighted in the work of Eck (1999).

$$H = \frac{PH + PHB}{9.81 * 1000} \tag{1}$$

$$AOD = AOD + EC55(t,V,lat,lon) * \Delta H(t,V,lat,lon) \tag{2}$$

WRF-Chem outputs variables in regular (hourly) time intervals, while AERONET keeps records in non-constant interval
steps. To temporally collocate the values, observations falling within a $\pm$30-minute range, centred at the model forecast hour, are averaged out. A spatial collocation is then implemented to modelled outputs of horizontal resolution 20 x 20 km. Modelled outputs are interpolated from a three-dimensional field to a horizontal plane at the station's height using the approach described in Ladwig (2017). In summary, the data are interpolated from a curvilinear grid to an unstructured grid using the nearest grid point to the station's coordinates. This approach has been repeated using the four and nine neighbouring grid points with
no statistically significant differences. Finally, an inverse distance squared algorithm is applied to output the variables at the coordinates and elevation of each station.



### 2.5.2 EMEP

The EMEP (European Monitoring and Evaluation Programme) is a pan-European database of ground-based aerosol concentration observations. EMEP was established following the convention of Long-range Transboundary Air Pollution in 1979.

European member states are legally bound to monitor and report emissions to EMEP with a standard temporal resolution of daily intervals. These are then stored in the EMEP open access database and are mapped using a 0.1° x 0.1° longitude-latitude grid (Pinterits, 2021).

To compensate for the poor temporal and spatial resolution of the EMEP database for the period under investigation, hourly products of PM10 at the EMEP background station of Agia Marina, Cyprus, were provided by the Department of Labour

Inspection Cyprus (DLI) (Labour Inspection , 2021). For the comparison of observed PM10 with simulated values, a spatial collocation was carried out in a similar manner to the one described in section 2.5.1. Since both simulated and observed values are reported at hourly intervals, no temporal collocation was required.

### 2.5.3 Polly<sup>XT</sup>

The Polly<sup>XT</sup> lidar monitors the vertical profiles of dust concentrations at the PANGEA observatory at Antikythera, Greece

(Marinou, 2017). The products are obtained through the application of the Earlinet SCC algorithm described in D'Amico (2015) and in the methodology presented by Mamouri (2017), and Ansmann (2019). The products are only available during cloud-free conditions. It is noted that the calculated dust mass concentrations have an uncertainty of 20-30% for a predominantly dust-dominated layer (Ansmann, 2019), while the uncertainty increases for a moderately dust-dominated layer and can reach up to 100% in layers with a small contribution of dust particles (Marinou, 2019).

The Polly<sup>XT</sup> products were subjected to a spatio-temporal collocation to enable a direct comparison with the simulated data. Observations are derived from 642 m up to 14 km above sea level, with a 60 m vertical resolution. For the vertical collocation of the two datasets, the Polly<sup>XT</sup> products were averaged to the model vertical bins to match the vertical resolution of the WRF-Chem model. Meanwhile, the same methodology described in section 2.5.1 has been used for the horizontal collocation. Polly<sup>XT</sup> products are derived in temporal averages between 30-minutes and 2-hours depending on weather limitations. Hence

for the temporal collocation, model hourly products were averaged to the Polly<sup>XT</sup> time windows.

### 2.5.4 MIDAS

For evaluating the WRF AOD spatial patterns, the recently developed MIDAS dataset (Gkikas, 2021, 2022) has been utilised as a reference. MIDAS provides columnar mid-visible (at 550 nm) dust optical depth (DOD) derived through the combination of quality-assured MODIS-Aqua AOD retrievals and the portion of AOD attributed to DOD (MDF; MERRA-2 dust fraction)

extracted from MERRA-2 (Modern-Era Retrospective analysis for Research and Applications version 2; (Gelaro, 2017; Randles, 2017)). Within MIDAS, the quality assured MODIS-Aqua AOD, as well as DOD (along with its associated uncertainties), are provided at a fine spatial resolution (0.1° x 0.1°) and at a global scale (both over land and maritime surfaces). Since the current WRF model version does not output DOD, the evaluation is focused only on the AOD simulated fields, mainly driven





by the spatiotemporal variations of the intense dust loads, spreading within the region of interest and dominating other aerosol
species.

In order to achieve the optimum MIDAS-WRF collocation, we are processing the swath level MIDAS data, where for the
constructed domain, MIDAS has approximately 5 to 7 daily overpasses corresponding to 5-minutes segments (Levy, 2013). The
segments have been reprojected from their native grid (Hubanks, 2018) to an equal latitude-longitude grid at 0.1° x 0.1° spatial
resolution. In contrast to MIDAS, WRF AOD is mapped on an equidistant, 20 x 20 km, Lambert conformal conic projection.
To project MIDAS and WRF-Chem AOD on a common grid, both have been regridded on an equal latitude-longitude grid at
0.4° spatial resolution. Regridding was carried out using the nearest neighbour method with a search radius around each grid
pixel set at 20 km. The radius was tested with values ranging from 10 to 50 km, resulting in percentage difference ranges of
0.008% - 0.04% and 0.0007% - 0.003% for MIDAS and WRF outputs, respectively. The availability of MIDAS observations
depends on the clouds' presence and deterring reasons for retrieving MODIS AOD. This leads to sparse AOD grid values
compared to the WRF continuous domain coverage. Thus, a mask function is applied on WRF in the areas where MIDAS
failed to resolve a value for AOD and simultaneously temporally collocates the two datasets. For the temporal collocation,
three methods have been tested: a 3-hour rolling average, a weighted average and use of the nearest hour. The latter was
performing poorly relative to the former two and was discarded, while a comparison between the two other methods revealed
negligible statistical differences. The weighted average method has been used to compute the collocation following Eq. 3,
where $AOD_i$ refers to the AOD at the nearest hour, $AOD_{i+1}$ to the AOD for the hour ahead and min to the minute of the MIDAS
overpass. The hourly outputs have then been summed to produce daily AOD maps. The edges of the overpasses have a slight
overlap; to overcome this, values at the overlaps were averaged. Finally, the daily sum maps were averaged to produce a single
map for the whole period under investigation.

$$AOD = AOD_{i+1} * \frac{min}{60} + AOD_i * (1 - \frac{min}{60}) \tag{3}$$

### 2.5.5 LIVAS

For the vertical assessment of the simulated dust patterns in the broader study region, the LIVAS (LIdar climatology of Vertical
Aerosol Structure for space-based lidar simulation studies) pure-dust product, initially presented in Amiridis (2013) and up-
dated in Marinou (2017), was utilised. LIVAS comprises a global dataset covering the period between 06/2006 and 12/2020 and
is provided on per-granule, level 2 resolution similar to the original CALIPSO (Cloud-Aerosol Lidar and Infrared Pathfinder
Satellite Observations) level 2 profile products.

The pure-dust extinction coefficient product has been developed through the application of the depolarisation-based sepa-
ration method introduced by Sugimoto (2003) and Shimizu (2004) and optimised for the Saharan region by Tesche (2009).
Marinou (2017) has calculated the uncertainty of the product in the region under investigation to be less than 20% for altitudes
up to 6 km. The products have a fine vertical resolution of 60 m and are projected on a uniform grid of 1° × 1° horizontal



resolution. To make observations directly comparable to simulated values, the WRF horizontal grid has been regridded to a 1°
x 1° lat-lon grid and vertically collocated following the methodology described in section 2.5.3.

## 3 Results and Discussion

In section 3.1, the WRF-Chem model runs for spring and autumn are compared, and the meteorological conditions and the
subsequent effects on dust transport for the selected study period are discussed. In sections, 3.2 and 3.3, comparisons of
assimilated and control outputs to ground- and satellite-based observations are summarised.

### 3.1 Seasonal Patterns and Dust Outbreak Oct. 2020

The WRF-Chem model, using both control and assimilated Aeolus ECMWF-IFS datasets, was run for two months in spring and
two in autumn. Fig. 1, depicting the differences (assimilated - control) of the averaged dust concentrations and wind vectors,
reveals that the use of the assimilated dataset has negligible differences to the control one during the spring months (1a), while a
difference is observed for autumn (1b). During the spring months, the differences in dust concentration between the two model
runs are less than 20 $\mu$g/m$^3$ for most of the study region. Meanwhile, for autumn, the differences are more pronounced. A
dipole seems to prevail, with the control run having higher concentrations over the Central Mediterranean and the assimilated
run over the Eastern Mediterranean, especially true for October. Additionally, a comparison of the two model runs using PM10
from the EMEP station of Agia Marina, Cyprus, for spring supports this finding (Fig. A2 of the Supplement). This finding
suggests that the impact of the assimilated dataset has temporal (seasonal) variation for the region under investigation, which
could be confirmed from long-term runs.





## Comparison of model runs for Spring and Autumn

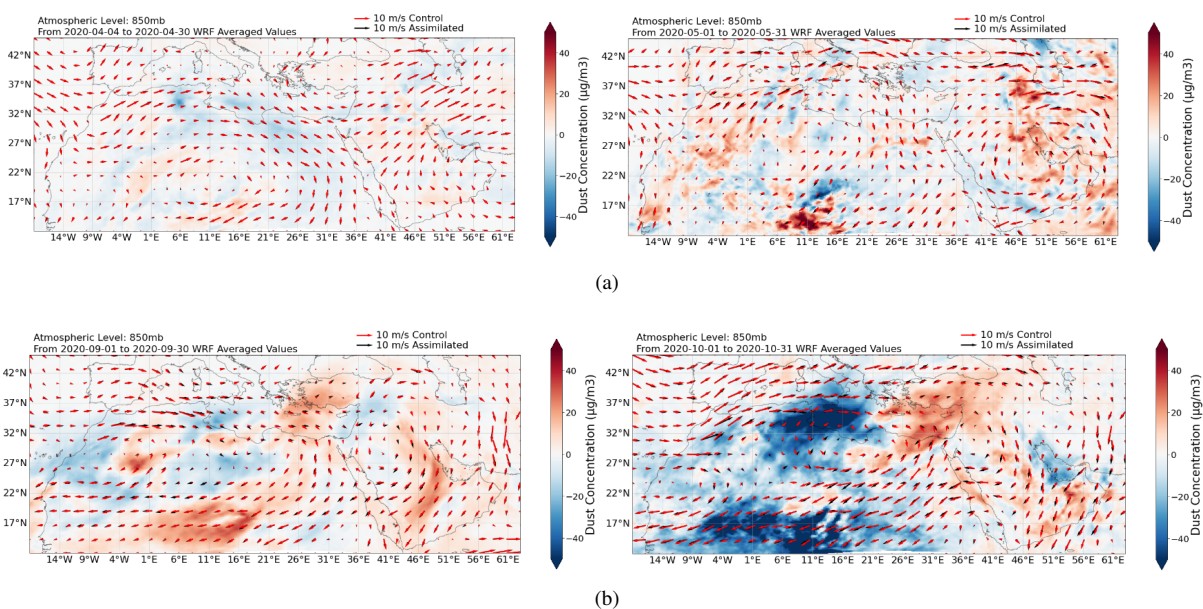

**Figure 1.** Depiction of the difference in average dust concentration and wind vector differences between the assimilated and control datasets, averaged for Spring months (1a) and Autumn months (1b) for the atmospheric pressure level of 850 mb, where the red arrows represent the control wind vectors and the black arrows the assimilated ones.

The availability of surface PM10 observations at the Agia Marina, Cyprus, station, allowed for an initial assessment of the two runs for autumn. Depicted in Fig. 2, two instances, highlighted in blue, mark periods where the assimilated run outperforms the control run, while one instance, highlighted in red, depicts the opposite. Summarised in Table 1 is the statistical analysis
for the autumn period, as well as for the three highlighted instances, where counts refer to the number of data points within each timeframe, r stands for the correlation coefficient and IOA for the Index of Agreement. The IOA measures the closeness of magnitude between two variables and is a unitless metric of range 0 to 1, where 1 indicates perfect agreement (Willmott, 2011). During most of the period, minor differences are recorded between the two runs. However, the assimilated run performs slightly better than the control, with the highest improvement observed for the interval of 20-25th of October. It is noted that
the total sample size is small to draw statistically significant conclusions. The period 14-25th of October 2020 was selected due to the formation of a dust storm of anticyclonic nature affecting the modelled domain, with notable deviations between the two model runs. The selected period allows for a thorough investigation of the impact of ECMWF-IFS Aeolus on meteorology, hence dust mobilisation.




**EMEP Timeseries for PM10 at Agia Marina, Cyprus, station for Autumn**

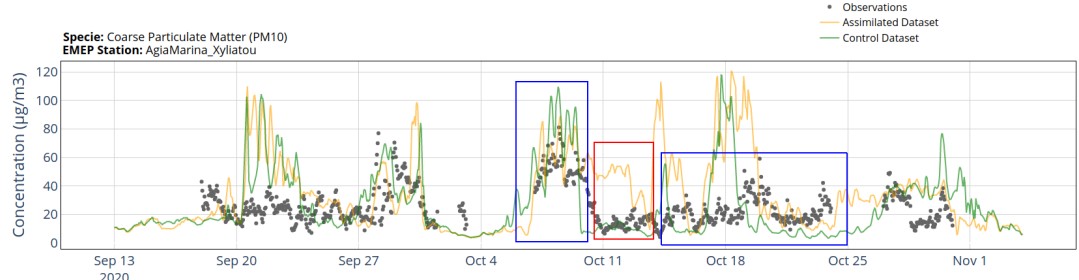

**Figure 2.** Timeseries of PM10 concentrations recorded at Agia Marina, Cyprus, ground station for the months of autumn, where the green trendline represents the control run, the yellow trendline the assimilated run and the black dots the observed values. Additionally, the blue highlighted boxes represent periods where the assimilated run outperforms the control run, while the red box represents the opposite.

**Table 1.** Statistical comparison of modelled runs with PM10 concentrations recorded at Agia Marina station, Cyprus, for the whole period and the highlighted time windows.

| Periods | Whole Period | | 07/10/2020 - 10/10/2020 | | 11/10/2020 - 14/10/2020 | | 14/10/2020 - 25/10/2020 | |
|---------|---------|------------|---------|------------|---------|------------|---------|------------|
| Counts | 952 (100%) | | 46 (4.8%) | | 94 (9.8%) | | 146 (15.3%) | |
| Dataset | Control | Assimilated | Control | Assimilated | Control | Assimilated | Control | Assimilated |
| r | 0.31 | 0.33 | 0.37 | 0.43 | -0.53 | 0.11 | -0.12 | 0.28 |
| IOA | 0.51 | 0.52 | 0.42 | 0.58 | 0.28 | 0.27 | 0.22 | 0.34 |

In the study of Hatzaki (2014), two major anticyclonic routes were identified, one parallel to the Iberian Peninsula and another parallel to the North African Coast, with the latter being the dominant route during summer and autumn. Consistent with past literature, during the 14-19th of October, a high-pressure cell develops in both simulations covering the Levantine basin and extending to North Africa, as seen in Fig. 3. The high-pressure conditions usher clockwise wind motion forming an anticyclone just north of the great sand sea desert in the Sahara region, 30° N and 25° E. The warm-core anticyclone develops from the convergence of the upper troposphere leading to air subsidence and warmer temperatures (Musk, 1988). The anticyclone develops near the 30° N line, consistent with past literature findings of warm-core anticyclonic development in the subtropics and midlatitude regions (Flocas, 2001; Hatzaki, 2014).

The anticyclogenesis on the 14-19th mobilises and transports dust through the gulf of Sidra into the Mediterranean basin. The two models have near-identical pressure zones for this period (Figs. 3a and 3b). However, the minor deviations lead to dust transported North-East in the assimilated run, while South-West in the control run, as depicted in Fig. 4a. This can be attributed to the extension of the high-pressure cell North-West of Cyprus in the assimilated run, driving winds towards the Dodecanese and Anatolia. In contrast, the weaker high-pressure cell in the control run leads to dust transport southwards from the Levantine basin to Egypt. During the 20-25th, the high-pressure system moves westwards. In the control run, it remains more defined (Fig. 3c), while in the assimilated run, it weakens and dissipates (Fig. 3d). A second anticyclone forms in the control run with its





focal point just off the coast of Tunisia, indicated in Fig. 4b and also observed in terms of absolute vorticity (see Fig. A3 of the
Supplement). The anticyclonic motion is not pronounced in the assimilated run, leading to markedly lower dust concentrations
over the Central Mediterranean and higher values over Egypt and the Middle East. The higher concentrations can be attributed
to the more stagnant conditions in the Eastern Mediterranean, where clockwise wind motion around 30° N 40° E mobilises
dust from the Arabian desert towards the Red Sea and the Levantine basin. The anticyclonic motion modelled in the control run
leads to higher wind speeds over the Central Mediterranean and Libya, while north-easterly winds pass through the Levantine
basin (Fig. A3a of the Supplement). Meanwhile, in the assimilated run, lower wind speeds are simulated over the Levantine
and Central Mediterranean basins, as well as on the coastline of North Africa (Fig A3b of the Supplement).

Additionally, FLEXPART-WRF air mass back-trajectories were simulated for a total of four, 5-day periods (14-19[th] and
20-25[th]), with particles arriving at the Antikythera, Greece station and the Agia Marina, Cyprus station. Similar air-mass routes
were noted for both stations with results being more notable for particles arriving at the Agia Marina station on the 19[th] of
October. Fig. 5 reveals apparent differences in the vertical height and source regions of aerosols arriving at the Agia Marina,
Cyprus station, on the 19th of October at 02:00 UTC. The assimilated run simulates the arrival of aerosols from North African
dust sources, while in the control run, aerosols originate in continental Europe.

**Pressure gradient of model runs for the 14-25[th] of October 2020**

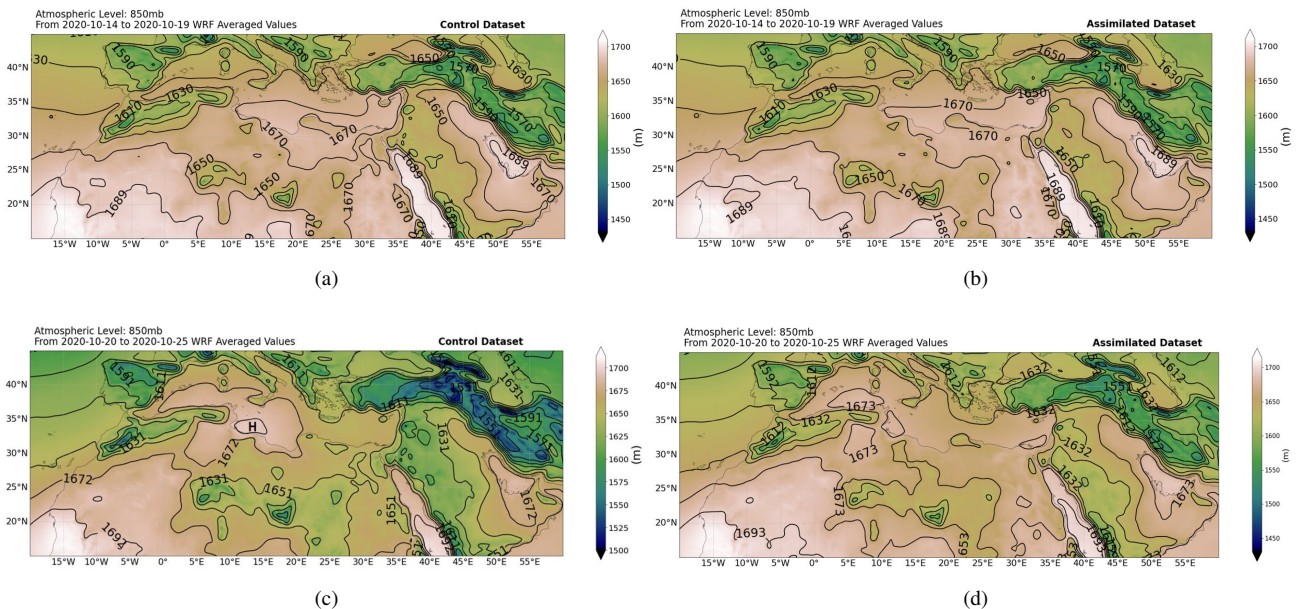

**Figure 3.** Representation of the averaged geopotential height for the 14-19[th] of October in the control run (3a) and assimilated run (3b) and
for the 20-25[th] of October in the control run (3c) and assimilated run (3d) for the atmospheric layer at 850 mb.



**Dust Concentration Differences between the model runs for the 14-25<sup>th</sup> of October 2020**

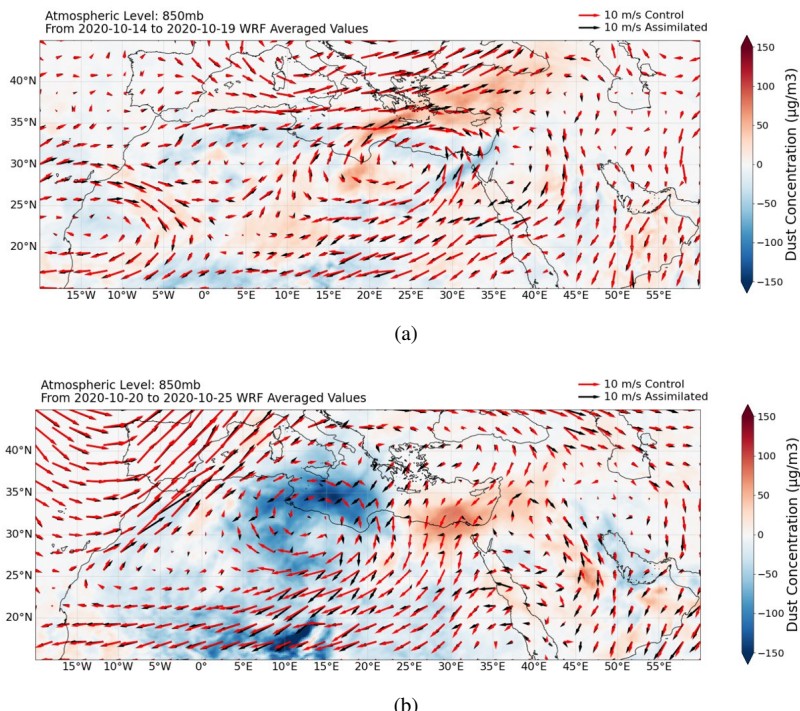

**Figure 4.** Averaged dust concentration differences (assimilated - control) and wind fields at the atmospheric level of 850 mb for the time-averaged periods; 14-19 of October (4a) and 20-25 of October (4b) where the red arrows represent the control wind vectors and the black arrows the assimilated ones.



**FLEXPART backward trajectories for the 19[th] of October 2020 at Agia Marina Cyprus**

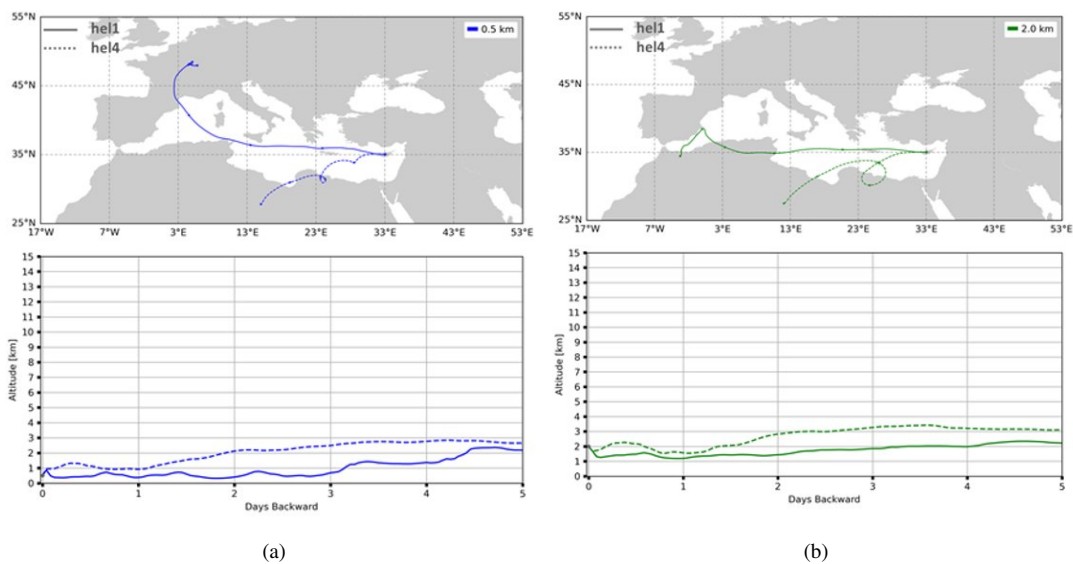

(a)                                                    (b)

**Figure 5.** The top panel figures depict the FLEXPART backward trajectories for tracer particles arriving at the Agia Marina, Cyprus station at 0.5 km (5a) and at 2 km (5b) on the 19[th] of October at 02:00 UTC. A solid line represents the trajectory of the control run, and a dashed line of the assimilated run. Additionally, the altitude inclinations of the particles are depicted in the bottom panel figures.

## 3.2 Ground-Based Evaluation

For the horizontal, spatio-temporal evaluation of the model runs, AOD from 56 AERONET sun-photometers within the whole
extended domain have been sourced. The map of stations used can be found in Fig. A4 of the Supplement. For the vertical evaluation of dust concentration, the Polly[XT] lidar at Antikythera station in Greece has been used.

In the study of Formenti (2001), the Mediterranean basin was characterised as a hotspot of long-range transport of tropospheric trace gases and aerosols. With base values 2 to 10 times higher than the hemispheric background troposphere (Lelieveld, 2002). The pressure gradient between the Azores high and Asian monsoon causes an eastward influx of small-sized particles
in the EMME region (Lelieveld, 2002). To identify dust-dominated loads of AERONET total atmospheric column retrievals, a filter on AOD and Ångström exponent needs to be applied. In previous AOD studies in the EMME region, the cut-off thresholds for AOD and Ångström indicating a dust-dominated AOD have been placed within the ranges of $> 0.15$-$0.35$ and $< 0.40$-$0.75$ respectively (Fotiadi, 2006; Basart, 2009; Toledano, 2007). The thresholds of AOD $> 0.15$ and Ångström exponent $< 0.75$ have been selected following the study of Gkikas (2021), which found these thresholds to be well performing in capturing
coarse particles. Thereby, values that satisfy this condition are classed as dust-dominated and are used in the statistical analysis summarised in Table 2. It is noted that the unfiltered AOD readings, accompanied by the Ångström exponent values, are used in the time-series plots (Fig. 6).





Statistical comparison of all 56 AERONET stations reduced the improvement obtained when implementing the Aeolus wind dataset relative to the comparison with selected stations. Metrics were produced for selected stations impacted by the anticyclogenesis mentioned above (14-25[th]), as well as for the two smaller time windows (14-19 and 20-25). This was done to test whether the improvement arises due to the materialisation of an anticyclone only in the control run from the 20[th] onwards (see 3.1). For both periods and the whole period, an improvement of 0.20-0.22 in the correlation coefficient is obtained using the assimilated dataset. This indicates that the improvement is not solely attained from an isolated instance. The anticyclonic conditions that prevail in the control run for the period of 20-25[th] of October 2020 over the Central Mediterranean are the underlying reason for the higher AOD profiles modelled at the impacted stations of; Lampedusa and Lecce University in Italy, Gozo in Malta and Finokalia and Antikythera in Greece. Meanwhile, the more stagnant wind conditions simulated in the assimilated run produced lower AOD values atop these stations and are more coherent to observations, as depicted in Fig 6. During this period, the assimilated run computes higher dust concentrations in the Levantine basin translating to a high AOD, consistent with a high AERONET AOD and low Ångström exponent recorded at Agia Marina and CUT, Cyprus stations, as well as Tel-Aviv University and Sede Boker stations in Israel.

**Table 2.** Statistical comparison of filtered AERONET AOD observations to model outputs for all 56 AERONET stations and for selected stations for the 14-25[th] of October 2020.

| | **All Stations** **14-25/10/2020** | | **Selected Stations** **14-25/10/2020** | | **Selected Stations** **14-19/10/2020** | | **Selected Stations** **20-25/10/2020** | |
|---|---|---|---|---|---|---|---|---|
| **Datasets** | Control | Assimilated | Control | Assimilated | Control | Assimilated | Control | Assimilated |
| **Counts** | 4835 (100%) | | 437 (9.0%) | | 212 (4.4%) | | 263 (5.2%) | |
| **r** | 0.46 | 0.51 | 0.12 | 0.33 | 0.23 | 0.43 | 0.22 | 0.44 |
| **IOA** | 0.41 | 0.45 | 0.38 | 0.52 | 0.48 | 0.59 | 0.37 | 0.52 |





## Model evaluation through AERONET AOD, October 2020

**Figure 6.** Timeseries plots of observed AOD (red points) from AERONET stations within the model domain, contrasted against assimilated AOD (yellow) and control AOD (green). Additionally, the black points represent the Ångström Coefficient obtained from the AERONET stations.

To supplement AERONET observations, the vertical dust profile at Antikythera station was obtained through the Polly[XT] lidar, depicted in Fig 7. In the works of Papayannis (2005), Mona (2006) and Kalivitis (2007), the highest dust concentrations in the Mediterranean during autumn were observed at 3-5 km, consistent with both model runs and Polly[XT] observations. During the 15-19[th] of October, the assimilated run overestimates while the control run underestimates dust concentrations, with the former having a better fit to the observed vertical structure. The over/under estimation could be attributed to the 20-




30% uncertainty of the lidar products (Ansmann, 2019). The formation of an anticyclone in the control run is materialised as a dust plume over Antikythera, arriving on the 20[th] and dispersing by the 22[nd] (Fig. 7a). In contrast, the assimilated run did not simulate the dust plume and is in better agreement with the limited observations available from Polly[XT] and the complementary observations from MIDAS and SEVIRI. Where for MIDAS, AOD values less than 0.3 are recorded in the area and from the

SEVIRI natural enhanced imagery and dust RGB composites, no dust plume is resolved in the area (see Figs. A5 and A6 of the Supplement). During the 23-25[th], the assimilated run simulates the arrival of a dust plume earlier than observed but has a more consistent vertical structure relative to the control run.

Overall, these results support that when implementing the assimilated dataset, the predictive ability of the WRF-Chem model for the specified regions is improved. This is perhaps related to the volatile conditions present in the Mediterranean during the

transitional autumn season.





**Figure 7.** Collocated vertical dust concentrations for the control run (7a), assimilated run (7b) and Polly[XT] observations (7c) at the Antikythera, Greece, station for the 15-25[th] of October 2020.



### 3.3 Satellite-Based Evaluation

MIDAS and LIVAS satellite observations were sourced to complement ground-based observations from AERONET and Polly$^{XT}$. MIDAS provides aerosol observations at wide spatial coverage with a fine spatial resolution, whereas LIVAS provides vertically resolved retrievals at fine vertical resolution.

A comparison of the model runs with observations from MIDAS is depicted in Figs. 8a and 8b, where the highlighted areas indicate locations of significant discrepancies between control and assimilated model runs. As expected, in both simulations areas of high AOD are concentrated at the Sahel, a dust source region active throughout the year with dust activity peaking in spring (Ravi, 2011; Middleton, 2001). The small Ångström exponent values positioned atop these areas confirm the presence of large dust aerosol particles. Meanwhile, high AOD values in North-Eastern Europe bear a higher Ångström exponent pointing
to smaller aerosols, likely of anthropogenic origin. Introducing the Aeolus assimilated wind dataset improves cohesion between simulated and observed values. This is especially true over the Central Mediterranean, where positive bias is reduced by 45%, attributed to the control run simulating an anticyclone not seen in the assimilated run. Furthermore, a reduction in negative bias by 8% is achieved over the Fertile Crescent (Mesopotamia), a dust-source region that became active from 2006 onwards due to prolonged periods of drought (Kelley, 2015).

Fig. 8c shows the observed AOD values averaged for the period 14-25$^{th}$ of October 2020 with three circled localities where the model, in both runs, underestimated AOD values. AOD present at locality 1 is most likely derived from the Bodélé depression, an enclosed basin of alluvium silts deflated by a low-level jet transporting dust south to the gulf of Guinea (Todd, 2007; Engelstaedter, 2007; Washington, 2005). The Bodélé depression has been the subject of various past literature due to the inability of global dust models with a coarse horizontal resolution to accurately depict AOD in the area (Bou, 2009; Huneeus, 2011;
Haustein, 2012). Locality 2 spans over the Mauritania-Mali source area, the Taoudenni basin, where the modelled, northerly winds displace aerosols further South for both runs overestimating AOD values relative to MIDAS. While locations 1 and 2 are dust-source regions strongly affecting areas in North Africa, the North Atlantic Ocean and the Mediterranean, location 3 is a critical dust-source influx for the EMME region. The underestimation of model AOD values in Mesopotamia has been related to the regime shift in dust inactivity from 2006 onwards (Notaro, 2015). It is apparent that the recent changes in land charac-
terisation are not reflected within the current model land configuration scheme. In future versions of the regional WRF-Chem model, the model-observation discrepancies could be further analysed, and appropriate re-configurations can be implemented to depict land use (change) accurately.

Using the LIVAS lidar allowed for the evaluation of vertical dust concentration as depicted by Fig. 9. In all instances, the dust resides up to 7 km with concentrations lower than 15 $\mu$g/m$^3$ at higher altitudes. This is consistent with past literature, which
determined the upper altitudes of dust in the Mediterranean to fall within the range of 5-9 km (Alpert, 2004; Papayannis, 2005). Revealed by Fig. 9a, just off the Libyan coast, the control run simulated a dust plume related to the pronounced anticyclonic activity, which is neither present in the assimilated run nor the observed dust vertical profile. Both model runs overestimated dust concentrations relative to LIVAS south-west from the Bodélé depression. The control run overestimated by approximately 500 $\mu$g/m$^3$ and the assimilated run by 200 $\mu$g/m$^3$, which could be attributed to the discussion regarding locality 1 (Fig 8c). For





the 21st of October (Fig. 9b), a dust plume is observed at the coast of Egypt, also modelled by the assimilated run, but shifted by 1 degree South whilst is absent in the control run.



(a)

(b)

(c)

**Figure 8.** Time averaged differences in AOD (WRF-MIDAS) for the control run (8a) and the assimilated run (8b), where the black boxes indicate regions of significant improvement when simulating conditions using the assimilated run. Additionally, 8c represents the observed MIDAS AOD values for the time-averaged period of 14-25th of October, where the highlighted red circles indicate locations where both runs underestimate AOD.




Only night-time profiles with a backscatter coefficient of more than 0.0008 and a height greater than 180 m above surface elevation were used to ensure a reliable statistical comparison with the LIVAS products. Additionally, the observations had to fit within the limits of a dust optical depth $\geq 0.01$ and $> 4$ cloud-free profiles falling within the one-degree grid-cell. Introducing

these filters saw both model runs recording an increase in correlation coefficient and index of agreement and a reduction in normalised mean bias (NMB) and root mean square error (RMSE). However, this resulted in a drop in the number of sample counts from 3,874 to 2,037, limiting the availability of comparable observations. Thus, a comparison was made for the whole domain rather than sub-regions, as conducted for MIDAS, and is summarised in Table 3.

It is apparent with both observation instruments that the introduction of the assimilated Aeolus wind fields improves the
predictive ability of the model by reducing positive biases and improving the correlation of modelled products to observations. Comparison with MIDAS observations saw a substantial reduction in positive bias by 44% in the Central Mediterranean region, accompanied by an increase in r by 0.19. This has been attributed to the control run simulating an anticyclone not present in the assimilated run. Over Mesopotamia, a reduction of negative bias of 17% is attained. To compensate for the small sample number in the regions mentioned above, a statistical analysis has been performed for a box spanning from 0° E, 30° N to 50°
E, 40° N, covering both the Central Mediterranean and Mesopotamia, which led to an increase in the sample pool. Within this area, significant improvements were recorded with an increase of r by 0.31, IOA by 0.18 and a reduction of positive bias by 23%. As observed in the statistical comparison with AERONET (section 3.2), comparing MIDAS observations to the whole domain reduces the improvements achieved using the assimilated dataset. This does not stand true for LIVAS, where data points are solely available atop CALIPSO-satellite tracks, and a synoptic comparison of all available overpasses improved r by
0.06, while reduced positive bias by 26% when using the assimilated products.





### Model evaluation through LIVAS vertical dust concentration for the 20<sup>th</sup> and 21<sup>st</sup> of October 2020

**Figure 9.** Collocated control and assimilated model runs to LIVAS depicting vertical dust concentrations for the dates; 2020-10-20-01:00 (9a) and 2020-10-21-00:00 (9b), along the CALIPSO tracks represented by the red lines in the horizontal AOD simulated values (9c, 9d), where for 9a and 9b the top panel represents the control run, the middle panel the assimilated run and the bottom panel LIVAS observations.





**Table 3.** Statistical comparison of modelled to observed products for MIDAS and LIVAS for the period 14-25[th] of October 2020.

| Region | Whole Domain | | 0° E, 30° N to 50° E, 40° N | | Mesopotamia | | Central Mediterranean | |
|---|---|---|---|---|---|---|---|---|
| **Comparison with MIDAS** | | | | | | | | |
| Datasets | Control | Assimilated | Control | Assimilated | Control | Assimilated | Control | Assimilated |
| Counts | 18532 (100%) | | 3443 (18.6%) | | 915 (4.9%) | | 1345 (7.3%) | |
| r | 0.52 | 0.54 | 0.32 | 0.63 | 0.49 | 0.60 | 0.53 | 0.72 |
| IOA | 0.71 | 0.72 | 0.53 | 0.71 | 0.52 | 0.64 | 0.70 | 0.81 |
| NMB | 57% | 52% | 52% | 29% | -41% | -34% | 68% | 24% |
| **Comparison with LIVAS** | | | | | | | | |
| Datasets | Control | Assimilated | Control | Assimilated | | | | |
| Counts | 2037 (100%) | | 325 (16%) | | | | | |
| r | 0.36 | 0.40 | -0.12 | 0.16 | — | | — | |
| RMSE | 90 | 68 | 82 | 63 | | | | |
| NMB | 48% | 22% | 26% | 22% | | | | |

## 4   Conclusions

The launch of the ESA's Aeolus satellite, acquiring wind profiles up to the lower stratosphere for the first time, was a significant step forward for Earth Observations. Aeolus filled a critical gap in the Global Observing System by providing vertically re-solved winds over remote oceanic and vast continental areas not well covered by conventional wind observations. Aligned with
the main scientific objective of the satellite mission, ECWMF, was the first meteorological centre that started the operational assimilation of Aeolus winds (January 2020), followed by other European institutes. Thanks to these activities, the beneficial impact of Aeolus winds on numerical weather prediction has been demonstrated via their assimilation into global atmospheric models. This advancement in the forecast models' predictive skills is anticipated to materialise also in aerosol and air-quality simulations, acknowledging the determinant role of meteorology on the processes governing the life cycle components of
airborne particles.

In the presented report, the effect on aeolian dust prediction from the inclusion of the ECMWF-IFS Aeolus assimilated Rayleigh-clear and Mie-cloudy wind products within the WRF-Chem model in spring and fall 2020 was studied. We focused on the broader Eastern Mediterranean and Middle East region, frequently affected by massive loads of mineral particles. The impact of the Aeolus dataset was more profound during the transitional October period compared to the spring months,
where the maximum difference in dust concentration over the study region was 20 $\mu$g/m$^3$ at the atmospheric layer of 850 hPa. Meanwhile, for October, distinct differences in dust concentrations were visualised, with values peaking over the Central Mediterranean in the control run and over the Levantine basin in the assimilated run. These suggested that the impact of the use of the assimilated dataset has temporal variation over the EMME region. Through further research, the observation regarding



the improved model predictability visualised only for the autumn months could be investigated to explain the driving forces
responsible for the improvement.

The period of 14 to 25 October 2020 has been investigated for the Eastern Mediterranean and Middle East region, where anticyclonic conditions prevail in the Central and East Mediterranean region leading to the transport of aeolian dust particles. The dust numerical model outputs were then evaluated against ground-based and satellite observations to ensure a complete and comprehensive assessment. Analysis of the October period revealed that in the control run, an anticyclone materialised
over the domain, which was much less pronounced in the assimilated run. Comparison of the model results to both ground- and satellite-based observations, including; EMEP, AERONET, PollyXT, MIDAS and LIVAS, allowed for a thorough investigation of numerical dust outputs both horizontally and vertically. In all cases, using the Aeolus assimilated wind products improved the model predictive ability with increases in correlation coefficient and index of agreement and decreases in positive and negative biases. The most significant improvements were observed when the statistical analyses were performed over the EMME region,
while comparisons with the whole simulated domain diffused the improvements. Specifically, for the period where a second anticyclone forms in the control run, the use of Aeolus resulted in a reduction of positive bias atop the anticyclone by 44% and an improvement in the correlation coefficient by 0.19. Through FLEXPART backwards-trajectory analysis, the source regions of aerosols were analysed. It was revealed that the control run had an influx of aerosols from continental Europe, while the assimilated run from the Saharan region. Concluding, incorporating the Aeolus products improves the predictive ability of the
WRF-Chem model for the East Mediterranean and Middle East regions by reducing positive bias and underestimates.

The report highlighted the importance of Aeolus in dust research and how it can serve as a benchmark for future relevant studies with emphasis on other natural aerosol species, such as the volcanic ash (Kampouri, 2022) and sea-salt. Relying on the same concept, a similar study (Gkikas, 2022) for the western Sahara, hosting some of the most active dust sources of the planet (Ginoux, 2012), and the Tropical Atlantic Ocean, receiving large amounts of mineral particles exported from the
Sahara Desert throughout the year, is in preparation. According to preliminary results in the framework of the JATAC campaign (Cape Verde, September 2021), there are evident modifications of the meteorological patterns. These are observed throughout the atmosphere, subsequently affecting the evolution of the Saharan dust plumes, which are more realistically represented in the numerical experiments initialised after assimilating Aeolus wind fields. An interesting aspect for future works would be a holistic approach for the assimilation experiments relying solely on Aeolus retrievals. More specifically, an investigation into
the feasibility of the assimilation of Aeolus wind profiles in regional atmospheric-dust models for producing meteorological and dust analyses could be carried out. These can be utilised to initialise short- to medium-term forecasts. Finally, a better representation of the simulated aerosol fields from Aeolus wind assimilation will result in an improved assessment of the aerosol-induced perturbations of the Earth-Atmosphere radiation budget with accompanying positive impacts on numerical weather prediction (Pérez, 2006; Gkikas, 2018; Benedetti, 2018). Thus, fulfilling the main scientific objective of the Aeolus
satellite mission.



# Appendix A

## A1

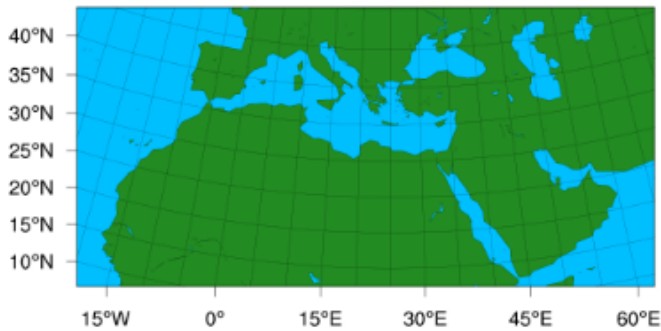

**Figure A1.** Model domain used in the WRF-Chem simulations.



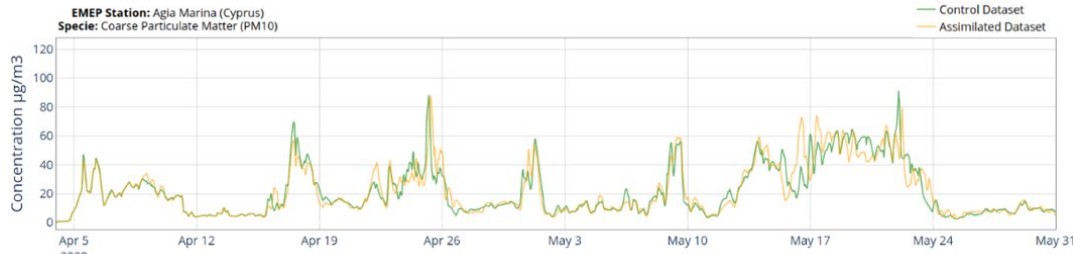

**Figure A2.** Comparison of PM10 at the EMEP Station at Ayia Marina, Cyprus, between the WRF-Chem simulations in Spring, where the green line represents the control run and the yellow the assimilated run.



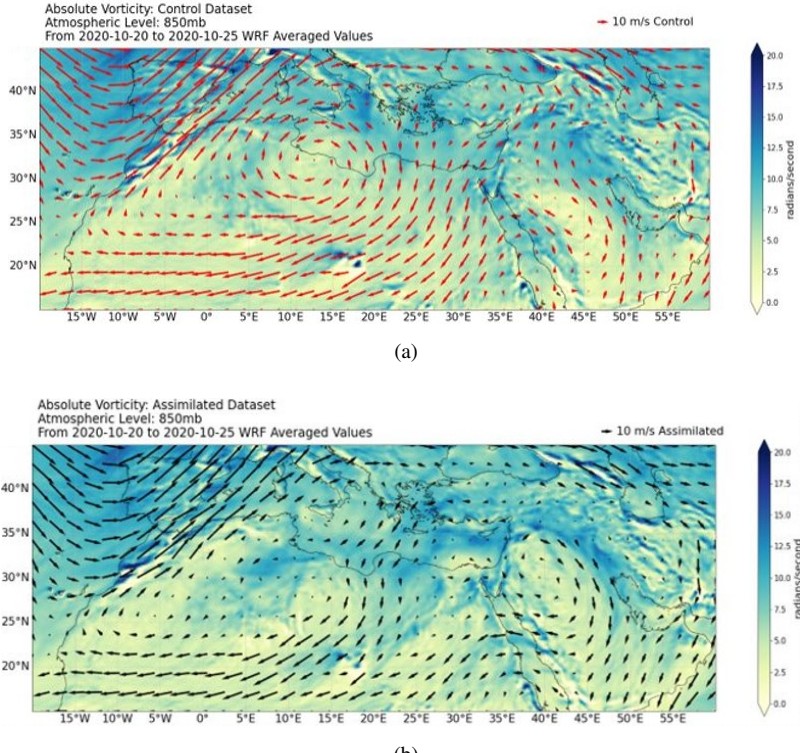

**Figure A3.** Comparison of model vorticity for control (A3a) and assimilated (A3b) runs (20-25 October 2020), where the wind vectors are represented in red for the control run and black for the assimilated run.





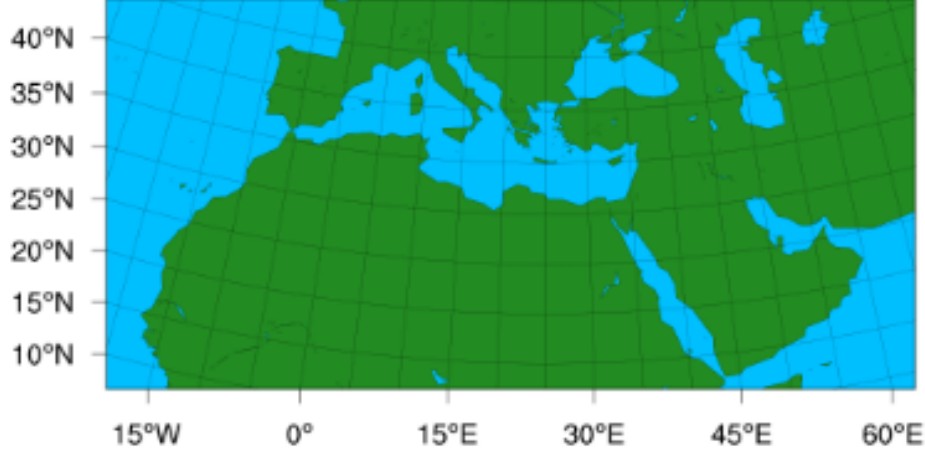

**Figure A4.** Map of all the AERONET stations within the constructed domain with available observations for the period of 18/09/2020 to 01/11/2020, used for the model evaluation.



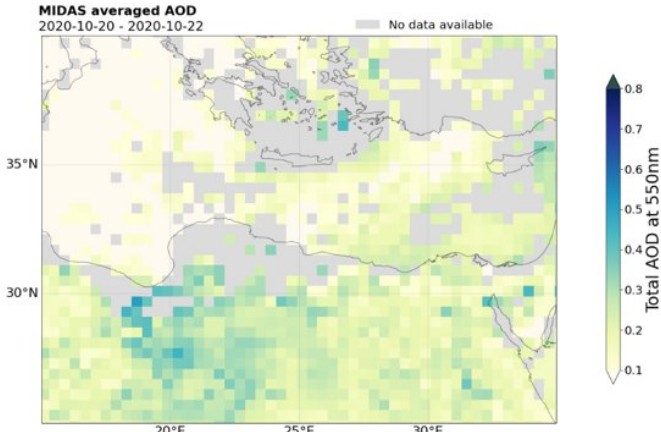

**Figure A5.** MIDAS AOD values for the time averaged period of 20-22[nd] of October 2020 for the model domain area surrounding the Polly[XT] lidar instrument.



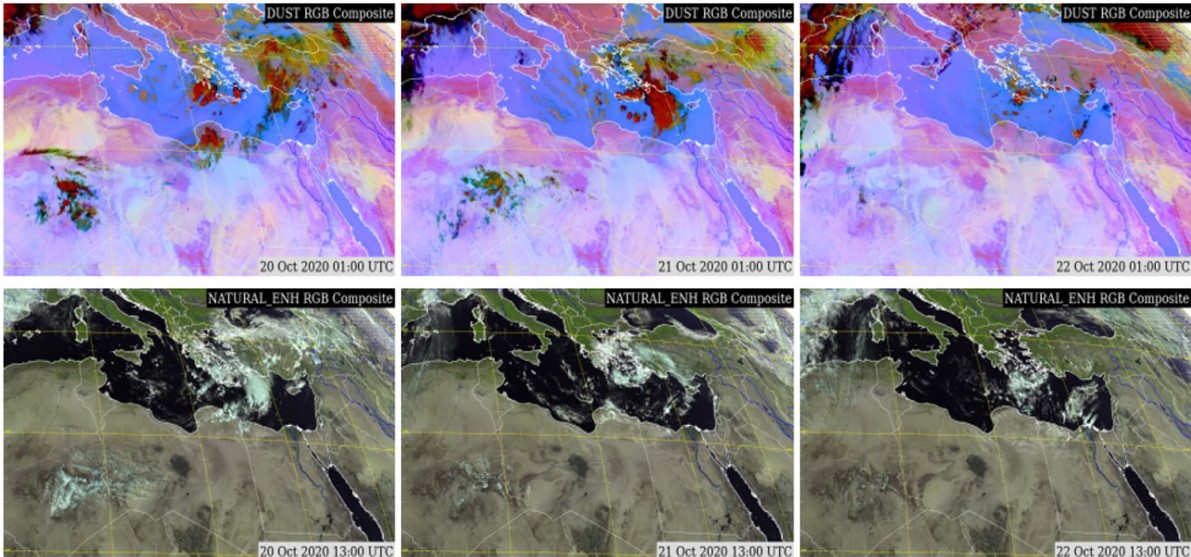

**Figure A6.** Representation of satellite imagery over Antikythera from the 20th to the 22nd of October, where in the top panel the Dust RGB composite colour is shown for night-time imagery at 01:00 UTC and in the bottom panel the natural enhanced imagery is shown for 13:00 UTC. The images of Fig. A6 are property of ©EUMETSAT [2020].



**Table A1.** WRF-Chem model configuration options used in the simulations.

| Process | Option | Reference |
|---|---|---|
| *Microphysics* | Morrison 2-moment scheme | Morrison (2005) |
| *Land-Surface* | NOAH Land Surface Model | Chen (2001) |
| *Boundary Layer* | Yonesi University (YSU) Planetary Boundary Layer | Hong (2006) |
| *Cumulus* | Grell 3D Ensemble Scheme | Grell (2002) |
| *Surface Layer* | MM5 Similarity Surface Layer Scheme | Zhang (1982) |
| *Radiation* | Rapid Radiative Transfer Model (RRTMG) | Iacono (2008) |
| *Gas Phase Chemistry* | Regional Atmospheric Chemistry Mechanism (RACM) | Stockwell (1997) |
| *Aerosols* | Model Aerosol Dynamics Model for Europe (MADE) Secondary Organic Aerosol Model (SORGAM) | Ackerman (1998) Schell (2001) |



*Author contributions.* Kiriakidis Pantelis: Produced the model experiments, Data Collection, Data Analysis and Writing of the Manuscript. Gkikas Antonis: Produced the model experiments, Data Collection, Contributed to the scientific discussion and Reviewed the Manuscript. Papangelis George: Produced the model experiments, Data Collection and Contributed to the scientific discussion. Christoudias Theodoros: Contributed to the scientific discussion and Reviewed the Manuscript. Kushta Jonilda: Contributed to the scientific discussion. Proestakis Emmanouil: Data Collection and Contributed to the scientific discussion. Kampouri Anna: Data Collection and Contributed to the scientific discussion. Marinou Eleni: Data Collection and Contributed to the scientific discussion. Drakaki Eleni: Contributed to the scientific discussion. Benedetti Angela: Contributed to the scientific discussion. Rennie Michael: Contributed to the scientific discussion. Retscher Christian: Contributed to the scientific discussion. Straume Anne Grete: Contributed to the scientific discussion. Dandocsi Alexandru: Contributed to the scientific discussion. Amiridis Vasilis: Contributed to the scientific discussion. Sciare Jean: Contributed to the scientific discussion.

*Competing interests.* The authors declare that they have no conflict of interest.

*Acknowledgements.* Gkikas A. acknowledges support by the Hellenic Foundation for Research and Innovation (H.F.R.I.) under the "2nd Call for H. F. R. I. Research Projects to support Post-Doctoral Researchers" (project acronym: ATLANTAS, project number: 544). The NEWTON project has been supported by ESA under Contract No. 4000133130/20/I-BG// Aeolus+ Innovation (Aeolus+I). The Cyprus Institute acknowledges support from the EMME-CARE project funded from the European Union's Horizon 2020 Research and Innovation Programme (under grant agreement no. 856612) and the Cyprus Government, and ACCEPT which is co-financed by the Norwegian Financial Mechanism (85 %) and the Republic of Cyprus (15 %) in the framework of the programming period 2014–2021.





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
