# Peer review of "The impact of using assimilated Aeolus wind data on regional WRF-Chem dust simulations"

_EGUsphere, 2022_

## Author Response (AR1)

The reviewer's comments are presented in black, while our answers are in blue. The references used to support our responses to the reviewer's comments are presented at the end of the text.

**Response to Reviewer 1**

**General comments:**

Review on paper titled "The impact of assimilating Aeolus wind data on regional Aeolian dust model simulations using WRF-Chem" by Pantelis Kiriakidis et al.

Paper compares output from two WRF-Chem model runs: with and without assimilated wind fields. In particular, authors use extensive set of ground and satellite observations to estimate model skill to simulate dust events. The article is well written and easy to follow. Authors conclude that they got significant improvements in dust simulation over the EMME region using assimilated wind data, while comparisons with the whole simulated domain diffused the improvements.

My main concern is the ability of WRF-Chem to correctly simulate the dust cycle in this study. Fig. 2 demonstrates it, i.e there is a better agreement in PM10 between runs rather than between runs and observations. See also Fig. A2, where good agreement between models is shown. Model also overpredicts PM10 and not capable to capture high pollution events in most of the cases. Thus, I think, authors pay attention to the 2nd order effect, while 1st order effect (dust simulation itself) is not satisfactory resolved. Therefore, I recommend revision before accepting for publication.

**Reply ->** We thank the referee for the constructive comments. Past dust simulation studies face various uncertainty sources in accurately depicting dust mobilisation and transport (Evan et al.,2014; Klose et al., 2014; Shao et al., 2011). This is also highlighted in the reference provided by the reviewer [2], where a source function was able to capture various point-scale dust sources, however, being applicable solely to high-resolution studies. Similarly, Nabavi (2017) tested a source function which was able to improve the predictive ability to the model, but signified deficiencies in the transport and deposition mechanisms used. Also stated in reference [2] and within our paper, past studies varied the parameterisation of the threshold velocity to improve the simulation of dust (Kok et al., 2014; Wu et al., 2016). Likewise, the paper aims to show the impact of using IFS assimilated Aeolus winds in the WRF-Chem model on dust transport, where the WRF-Chem model configuration follows the most recent parameterisations, source functions and mechanisms sourced from literature on dust simulation in the East Mediterranean and Middle East region. Thereby, using the most updated version of the model relative to dust simulations, we aim to show whether an improvement can be attained from incorporating the Aeolus wind fields.

Regarding the comparison with the EMEP dataset (Fig. 2), the authors demonstrate the deviations between the two model runs in autumn compared to the near-identical simulated PM10 values in spring, shown in Fig. A2. Furthermore, the highlighted instances in

Fig. 2 point out that strong deviations between model runs materialise in periods of anticyclogenesis, where statistical improvements achieved in the assimilated run can be seen in Table 1. Use of AERONET, Polly[XT], MIDAS and LIVAS, ground and satellite observation datasets, the paper aimed to observe the impact of Aeolus on horizontal, vertical and temporal evolution of dust in the region for the case study of 14-25 of October. Through statistical comparisons of the simulated values to observed ones, the assimilated run proved to be more able to accurately depict dust transport in the region.

**Specific comments**

- the title is a misleading. It is not clear, whether WRF model itself assimilates wind data or not.

  **Correction ->** The impact of using assimilated Aeolus wind data on Regional WRF-Chem dust simulations.

- Introduction is lengthy, 2nd paragraph on page 1 and 1st on page 3 could be shortened. References [2] and [3] devoted to rigorous dust simulation on the Middle East are missing in the manuscript.

  **Correction ->** Removed some of the context from paragraphs 2 and 3 of the introduction and included references 2. and 3.

- Not clear why authors used complex (MADE/SORGAM) aerosol scheme to simulate dust? Is there any justification for it? However, here [1] you may find some useful details on how to simulate dust in WRF-Chem using modal aerosol scheme.

  **Reply ->** We would like to thank the referee for the references which we have incorporated in the manuscript. We are using a modal aerosol scheme that was excessively tested and was shown to perform well across the EMME region [Georgiou (2018), Kushta (2018)]. This will allow us in the future to investigate the impact of using assimilated data on other components of the Earth system.

**Technical corrections:**

- Line 62: first occurrence of WRF-Chem. Please add reference.

  **Corrected**

- Line 90: HSRL, HLOS unknown abbreviations.

  **Corrected**

- Line 107: .. seasons of the region. Please specify which region.

  **Corrected**

- Line 130: Natural emissions. Please explain, what do you mean?

**Correction->** Changed to: Mineral dust and sea-salt emissions were calculated on-line by the WRF-Chem model, driven by IFS Aeolus-assimilated data

- Line 170-172: what type of FDDA you used? Not clear, who lateral boundary conditions can be improved by FDDA? If FDDA is enabled in WRF, then model fields (not observations) are nudged to reanalysis fields.

**Correction->** Following the reviewers comment we have modified the text to clarify as follows: FDDA has been applied towards IFS re-analysis fields with and without assimilated Aeolus observations. It was shown that by nudging above and within the Planetary Boundary Layer, the accuracy of the meteorological variables simulated within the WRF-Chem model is improved (Deng et al., 2007) and has since been used in other dust-related studies (e.g. Kumar et al., (2014)). Following this, the horizontal wind components, temperature and moisture were nudged in all the model vertical layers, except the surface level, with a nudging coefficient of $3 \times 10^{-4}\,s^{-1}$. Nudging was carried out at each time step throughout the whole simulation, with a time interval between analysis times of 6 hours. Ramping started at the last analysis time and ended as a step function.

- Line 188: remove ; ?

**Corrected**

- Line 200: height of (t,V,lat,lon) model level, and ΔH - width of the (t,V,lat,lon) model level.

**Corrected**

- Lines 200-201: Please remove in in units.

**Corrected**

- Formula 1: add (t,V,lat,lon) to PH and PHB

**Corrected**

- Formula 2: Replace by AOD(t,lat,lon)=∑EC55(t,V,lat,lon) ∗ ΔH(t,V,lat,lon), where ∑ - sum over V.

**Corrected**

- Figure 1: Land contours are hardly seen (Fig. A3 same).

**Corrected**

- Line 301: missing formula for IOA.

**Corrected**

- Line 317: 14-19th. Please add October.

**Corrected**

- Figure 5: what os hel1, 4 on plot legend? Please cut top altitude to 5km.

  **Corrected**

- Figure 6: Please replace AERONET-alpha by "Ångström Coefficient", replace y-axis label by "Ångström Coefficient"

  **Corrected**

- Figure 8: Please replace 'black boxes' by 'black rectangles'.

  **Corrected**

- Figure A1: Please move it to the main text and illustrate all (if possible) geographical locations mentioned in the study. Also plot locations of AERONET stations (remove Fig. 4a, which is empty anyway).

  **Corrected ->** Fig. A4 has now been corrected and includes the geographical locations mentioned in the study. Similarly, fig. A1 now includes geographical locations mentioned in the text, however, it was decided that fig. A1 will remain in the appendix.

**References given by the reviewer:**

1. Osipov et al, Severe atmospheric pollution in the Middle East is attributable to anthropogenic sources

2. Parajuli et al, Dust Emission Modeling Using a New High-Resolution Dust Source Function in WRF-Chem With Implications for Air Quality

3. Ukhov, A. et al. Assessment of natural and anthropogenic aerosol air pollution in the Middle East using MERRA-2, CAMS data assimilation products, and high-resolution WRF-Chem model simulations

**References used in the reply**

Evan, A. T., Flamant, C., Fiedler, S., & Doherty, O. (2014). An analysis of aeolian dust in climate models.Geophysical Research Letters,41,5996–6001. https://doi.org/10.1002/2014GL060545

Nabavi, S.O., Haimberger, L. and Samimi, C., 2017. Sensitivity of WRF-chem predictions to dust source function specification in West Asia. Aeolian research, 24, pp.115-131., Vancouver,

Klose, M., Shao, Y., Li, X., Zhang, H., Ishizuka, M., Mikami, M., & Leys, J. F. (2014). Further development of a parameterization for con-vective turbulent dust emission and evaluation based onfield observations.Journal of Geophysical Research: Atmospheres,119,10,441–10,457. https://doi.org/10.1002/2014JD021688

Kok, J. F., Albani, S., Mahowald, N. M., & Ward, D. S. (2014). An improved dust emission model—Part 2: Evaluation in the CommunityEarth System Model, with implications for the use of dust source functions.Atmospheric Chemistry and Physics,14(23), 13,043–13,061.https://doi.org/10.5194/acp-14-13043-2014

Shao, Y., Wyrwoll, K. H., Chappell, A., Huang, J., Lin, Z., McTainsh, G. H., et al. (2011). Dust cycle: An emerging core theme in Earthsystem science.Aeolian Research,2(4), 181–204. https://doi.org/10.1016/j.aeolia.2011.02.001

Wu, C., Lin, Z., He, J., Zhang, M., Liu, X., Zhang, R., & Brown, H. (2016). A process-oriented evaluation of dust emission parameterizationsin CESM: Simulation of a typical severe dust storm in East Asia.Journal of Advances in Modeling Earth Systems,8, 1432–1452.https://doi.org/10.1002/2016MS000723

**Response to Reviewer 2**

***General comments:***

This study aims to demonstrate that the assimilation of Aeolus in the model used as boundary conditions for the dust simulations improves the capability of the regional model to resolve dust loads.

The paper is readable and the results are validated from external measurements.

The study is based on several months of simulation but only focuses on a reduced period and one particular event. Studies with Aeolus data on other specific events (e.g., tropical cyclones) acknowledge that it is difficult to work on such a reduced set of events (see, for instance, DOI: 10.1002/qj.4370, The characterization and impact of Aeolus wind profile observations in NOAA's regional tropical cyclone model (HWRF) by Marinescu et al., 2022). The impact is not systematically in the direction of the average impact (i.e., assimilating Aeolus data can also be very detrimental in some cases).

I think this paper requires such a discussion on the significance of the results when considering the reduced set of events.

Without this discussion, this paper is still an important contribution to the demonstration of the usefulness of Aeolus data, through a well-documented case study. However, I don't think it stands as a solid proof by itself.

**Reply ->** We would first like to thank the reviewer for their time and constructive feedback. The paper does not aim to solely and uniquely evaluate the potential of assimilation throughout the lifetime of Aeolus but rather stands as a test case and proof-of-concept, and this is noted throughout lines 480-484 where various published papers and ongoing studies evaluating the performance of Aeolus data are mentioned. To correctly depict the significance of the paper the following sentence has been changed from "Concluding, the incorporation of the Aeolus products improves the predictive ability of the WRF-Chem

model for the East Mediterranean and Middle East regions, by reducing positive bias and underestimates." to "The benefits attained from the incorporation of Aelous, solely regard the period of 14 to 25 October 2020, where anticyclonic conditions prevail in the EMME and Central Mediterranean regions. Even though the period of improvement is statistically negligible compared to longer timescales, the strong reductions in positive bias and underestimates highlight the importance of Aeolus in further dust research.".

***Specific comments:***

- I do not understand the expression "comparisons with the whole simulated domain diffused the improvements" (l. 355) or "Statistical comparison of all 56 AERONET stations within the extended model domain diffuses the improvement" (l.436) and another occurrence on l. 468. In particular, this use of the word "diffuse". This might be a specific jargon that I am not aware of, but could you explain this notion with a different wording?

  **Corrected ->** Removed the use of the word diffuse throughout the manuscript

- l.88 Is "turmoiled" necessary?

  **Reply ->** Turmoiled was placed to demonstrate the effect political instability has in the world of academia, but has now been removed as requested.

- l.109: Did you mean "in section 2.1 to 2.5"?

  **Reply ->** Corrected to clearly reflect what is described per section.

- l.197 "miscrophysical"

  **Corrected**

- l.260 "deterrent", not sure I understand, did you mean "inherent"?

  **Corrected**

- l.263: why was the nearest hour so different from a 3-hour average? Is the model AOD noisy or extremely variable?

  **Reply ->** In short, the anticyclogenesis present during the study period would see a highly variable AOD. There are 3 to 5 MIDAS overpasses with a 5-minute gap between them. The closest hour approach is configured to compare MIDAS values that fall within hh:00 to hh:29 minutes to the WRF interval hh, where hh refers to the hour. For values at hh:30 to hh:59 the WRF interval of hh+1 is used in the comparison. Hence, if MIDAS had an overpass during 13:25, 13:30 and 13:35, the nearest hour approach will solely compare the 13:25 interval to the WRF product at 13:00 and the other two to the WRF product at 14:00. Thus, the rolling 3-hour approach would be more adept to capture and compare this variability relative to the nearest hour approach. Additionally, the 3-hour approach is more aligned with the weighted average approach, which uses two model hours and weights them according to whether the overpass was after hh:30 or before hh:30 (Eq. 3).

- l.265-269: that's a lot of averaging. How different are the values where there is an overlap for instance? In addition, I don't understand what is being produced here. Maps of MODIS AOD?

  **Reply ->** There are a total of 2,192, 5-minute, MIDAS retrievals that do not continuously cover the whole domain. To ensure all the available observations were used, daily sums were created with overlaps averaged. Then the daily sums were averaged to produce a single AOD map, being the most effective way of qualitatively visualising the comparison. Stated on lines 272 and 274 AOD maps comparing the collocated products were produced.

- l.295: Please describe better figure A2: It shows that both model runs give very similar results at the Agia Marina station during the spring period.

  **Correction ->** Clarified what figure A2 points at.

- l.305-306: statistical significance again… This contradicts the l.308 statement of a "thorough investigation"

  **Reply ->** Following lines 303-308 the statistical significance refers to the comparison with EMEP. While line 310 foreshadows the thorough investigation to follow for the stated period. Changes have been made to remove any contradiction.

- Fig. 5: I would suggest to either remove hel1 and hel4 from the legend or introduce it somewhere in the text.

  **Corrected**

- l.334: Isn't it 4 FLEXPART runs but only "two, 5-day periods"?

  **Corrected**

- l.342: Fig A4 does not show the AERONET stations

  **Corrected**

- l.411: LIVAS is a dataset, not a "lidar"

  **Corrected**

- Fig 8: Is it possible that dust events happening close to the domain boundary are less well resolved? (e.g. dust could be transported from outside the domain, across the boundary). There are also some discrepancies to the East of the Caspian Sea for instance. The other hypothesis would be that getting the magnitude wrong on strong events already produces a large error. And unfortunately, they happen close to the domain border for this period.

  **Reply ->** Dust events close to the domain boundaries are less resolved, in particular, discrepancies East of the Caspian Sea are noted. The relative error of simulating events increases with the magnitude of the event, hence strong events close to the boundaries inherit a larger error relative to events in the inner domain, which is the

case for this study period. This is now included in the text. Additionally, referring to Fig. 8c, location 1 the Bodele depression, has been the subject of various past model simulations identifying an inability of various models to accurately depict dust mobilisation in the locality, also included in the text.

**Response to Reviewer 3**

**General comments**

The authors have studied the impact of using Aeolus-improved meteorology initial and boundary conditions in WRF-Chem regional dust simulations. More specifically, the initial and boundary conditions are IFS fields obtained with the assimilation of Aeolus wind profiles that go to improve meteorological patterns and therefore dust transport. The paper is well written and structured with a clear scope and meaningful results. I have some questions for the authors and also comments that could help improving the paper.

We would like to thank the reviewer for their valuable feedback and comments which we feel they have helped improve the quality of the manuscript. Please find below our comments and changes. Thank you for the constructive feedback!

**Specific comments**

- I find the title misleading. It reads as if Aeolus wind data were assimilated in WRF-Chem. This is not the case. These data are assimilated in the IFS and IFS outputs are used as initial and boundary conditions of WRF-Chem. I strongly suggest to change the title to reflect more faithfully the work done.

  Correction -> The impact of using assimilated Aeolus wind data on Regional WRF-Chem dust simulations

- similarly, I suggest to rephrase the many sentences in the paper that describe the experiments done as data assimilation experiments, which, in my opinion, is not strictly correct (and unfair toward the amount of work that went into assimilating Aeolus in the IFS), unless I'm misunderstanding the simulations. If the latter, it would be good to clarify how the Aelous data are used in WRF. In my understanding IFS analyses of wind, temperature and moisture (with or without the assimilation of Aeolus wind profiles in the IFS) are used as boundary and initial conditions of WRF through nudging.

  Correction -> Changed the sentences referring to the assimilated Aeolus data to clearly reflect the work of the IFS and avoid confusion

- it would be good to provide more details of the nudging performed: frequency of nudging, nudging time scale, ramping period (if those apply).

  Correction -> The following has been added in the text: "the horizontal wind components, temperature and moisture were nudged in all the model vertical layers,

except the surface level, with a nudging coefficient of $3 \times 10^{-4}\,s^{-1}$. Nudging was carried out at each time step throughout the whole simulation, with a time interval between analysis times of 6 hours. Ramping started at the last analysis time and ended as a step function."

- please clarify the calculation of aerosol optical depth and the definition of some of the variables used. "total-column atmospheric extinction coefficient at the wavelength of 550 nm (EC55)": is not a profile instead of being total column (in eq 1 has in fact the dimension vertical layer)? "EC55 can be used as a proxy for dust optical depth": not really a "proxy", one is a profile, if I'm correct, the other is column-integrated, one refers to all types of aerosol and the other only to dust. Also, please review eq 1 (it is a line of code that must have been inside loops) so that is expressed well mathematically. Also, which are the optical properties and assumptions on particle size and shape used to derive EC55 from mass concentrations?

  **Correction->** Changes were made to equations 1 and 2. The explanation of EC55 was re-written to correctly reflect the variable. The following has been to the text to describe optical properties and particle size assumptions: Dust particles are assumed to be spherical and internally mixed in five differing size bins of an effective particle radius of 0.5, 1.4, 2.4, 4.5 and 8.0 µm. The Mie theory has been used to calculate the optical properties as a function of wavelength at each model grid point for the wavelengths of 300, 400, 600 and 999 nm as described in Barnard et al., (2010). More specifically, the aerosol optical thickness, the single scattering albedo and the asymmetry parameter were determined by interpolation at the wavelength of the centre of the band located between the four wavelengths (Chaibou et al.,2020).

- the flexpart part could be removed. What does add to the study?

  **Reply ->** We consider the use of the FLEXPART model in order to study the origin of particles at the Agia Marina, background station in Cyprus. Using FLEXPART we were able to distinguish that in the control experiment, particles arrived from Europe, while in the assimilated run the particles arrived from North Africa, consistent with observations.

- it seems that the main impact you see is in the transport of dust, why do you think you did not see a greater impact on the mobilization and emission fluxes? could you comment more on this aspect please.

  **Reply ->** As can be seen in the two animations now uploaded in the electronic supplement material, surface winds over dust sources have small deviations between the IFS with and without assimilation. Wind speed differences are more pronounced at the atmospheric layer of 850hPa, which led to the differences observed in the transport of dust rather than mobilisation and emission fluxes.

- sec 2.1 should not the scaling proportionality constant for the dust emission flux be unitless?

**Reply ->** Following the works of Ginoux (2001) and Zhao (2010) the proportionality constant is expressed in $\mu g s^2/m^5$

- sec 2.2 please review the paragraph, overall it does not read too well. For example, talking here about the "the control and assimilated runs" might confuse the reader since the same names are used for the WRF simulations; "2B10 baseline" maybe not all the readers would be familiar with this; "Rennie (2021) who provided the configuration" does not read well. Also, could be useful to specify more details: which IFS output is used by WRF, analyses or forecasts, and at which spatial and temporal resolution.

  **Correction ->** The IFS outputs utilized as initial/boundary conditions in WRF are exactly the same with those described in Rennie et al. (2021) (Section 2.3: Observing system experiments). More specifically, we are using the analysis IFS numerical outputs, which are produced every 6 hours and are projected at an equal lat-lon grid at 12 km x 12 km spatial resolution. Incorporated in the text and changes were made throughout section 2.2 so as to make it easier to read through.

- throughout the paper there is a bit of inconsistency among the length of the experiments. It would be good to clarify the dates. In the introduction: "April - May and September – November 2020" in 3.1: "was run for two months in spring and two in autumn". In 2.3: "periods 2020/03/04 - 2020/05/31 and 2020/09/01 – 2020/11/04". Btw, why March is not analyzed? Also, is a spin-up period taken into account for both seasons?

  **Correction ->** The 2020/03/04 was a typo, 2020/04/04 is the correct period, hence March was not part of the analyses. A spin-up period of 7 days is taken into account for both runs.

- sec 2.5.4: why both MIDAS and WRF-Chem AOD have been regridded rather than regridding only the finer resolution one? I'm not suggesting to redo things, just to understand the reason for a final overall coarser spatial resolution of 0.4.

  **Reply->** One of the main elements of the MIDAS dataset is its fine resolution. However, we have noticed on the AOD spatial distributions that, in some cases, there is "noise" when weak loads are recorded. After performing several sensitivity tests, we have decided to regird the raw MIDAS data from 0.1 x 0.1 to 0.4 x 0.4 degrees as a trade-off for minimizing the noise effect. Now included in the text.

- line 263: "the latter was performing poorly" relatively to what metrics?

  **Correction ->** The three approaches were compared to each other (relative difference). The one performing poorly had the highest difference from the other two. To clearly reflect this the line has been rephrased to "The relative difference between the latter and the former two approaches indicated poor performance and was hence discarded. Meanwhile, a comparison between the two remaining methods revealed negligible statistical differences." (Lines 268-270)

- Figure 4 does not contain the stations. Additionally, please consider reporting here or in A1 the main geographical names of places used in the discussions throughout the paper, and also the lidar station location.

  **Corrected**

- sec 3.2 Not clear why the AERONET AODs are filtered for dust-dominated conditions when the model output is AOD and not DOD (line 248: "current WRF model version does not output DOD")

  **Correction ->** The high dust concentration simulated over the Central and East Mediterranean allowed for the assumption of a dust-dominated AOD. Hence, we filter AERONET AOD observations in an effort to compare dust-dominated AOD columns. This explanation has now been included in the paper.

- Fig 6: is only AE>=0.75 shown? Can you please state it clearly. Also, the yellow color is difficult to see and to distinguish from the red.

  **Correction ->** On line 358 the following is stated "It is noted that the unfiltered AOD readings, accompanied by the Ångström exponent values, are used in the time-series plots (Fig. 6)." To make it clearer it has been rephrased to "It is noted that the unfiltered AOD readings, accompanied by the Ångström exponent values, are used in the time-series plots (Fig. 6), showcasing all available observations."

- 368: the plots show high concentrations also lower than that, at 1 km. Similarly line 410, I don't see the statement being consistent with the plot.

  **Correction ->** Revised the reference value to the correct one and rephrased the sentences to being consistent.

- 388-390 where is all this about AE shown?

  **Correction ->** Added a figure of AE captured by MIDAS in the Supplement, however after co-authors remark on the reliability of the AE measurements from MIDAS, these lines and the figure have been removed.

- Table 3: define somewhere the regions considered here. Line 423 says that no sub-regions are considered, but there is one considered beside the whole domain.

  **Correction ->** The second column for LIVAS refers to the boxed area of 0◦ E, 30◦ N to 50◦ E, 40◦ N and it has now been mentioned in the discussion (Line 443)

- Lastly, it would be fair to add in the conclusion that the results are relative to a case study only, basically a part of the month of October makes the main difference, and that they are not necessarily statistically significant

  **Correction->** Changed the conclusion accordingly and added the following "The benefits attained from the incorporation of Aelous, solely regard the period of 14 to 25 October 2020, where anticyclonic conditions prevail in the EMME and Central Mediterranean regions. Even though the period of improvement is statistically

negligible compared to longer timescales, the strong reductions in positive bias and underestimates highlight the importance of Aeolus in further dust research."

**Technical corrections**

- 24: disease → diseases

  **Corrected**

- 85: please review the whole sentence: "an observational coverage network able to feed the model" does not make proper sense as it is written.

  **Correction ->** Changed to: The model predictive ability also benefits from an observational coverage network able to provide the model

- 88 combated → overcome?

  **Corrected**

- 104: "the ECMWF-IFS assimilated Aeolus wind fields provided by ESA are implemented in the WRF-Chem" is not written correctly, please rephrase it. ESA provides the wind fields, these are assimilated in the IFS and then used in WRF-Chem.

  **Correction ->** Changed to: In this study, the Aeolus wind fields provided by ESA and assimilated by ECMWF-IFS are implemented in the WRF-Chem

- 106: in consideration → in the inclusion or not

  **Corrected**

- 109: remove or substitute "Following", also in line 151

  **Corrected**

- 114: "the incorporation of the Aeolus assimilated wind fields within the ECMWF-IFS datasets" could be phrase it better. The wind fields are assimilated in the IFS model.

  **Corrected**

- 120: "adjusted": in which respect?

  **Corrected**

- 122 mean bias does not need to be capitalized

  **Corrected**

- 126 EDGAR-HTAP: use first the full explicit name and then the acronym. Please check also other similar occurrences.

  **Corrected**

- 136: please check the grammar or the word "treatment" here: "was consistent with [ … ] aerosol size treatments"

  **Corrected** -> Removed treatment

- 140: "more accurate": please specify more than what.

  **Corrected**

- 178: "the assimilated ECMWF-IFS dataset" is not strictly "assimilated"

  **Corrected**

- 194: "measure": they are rather retrievals from measurements

  **Corrected**

- 200: "total"?

  **Corrected**

- 237: "model vertical bins"?

  **Corrected**

- 286: Oct. → in October

  **Corrected**

- 291: specify that these statements are for the pressure level considered.

  **Corrected**

- 296: "could be confirmed from long-term runs" → should be confirmed by

  **Corrected**

- Figure 1: Spring and Autumn don't need to be capitalized, caption and title, please

  **Corrected**

-  check the rest of the paper

  **Corrected**

- Figure 1 in average → in monthly average

  **Corrected**

- Figure 1: "and wind vector differences" → and of wind vectors (are not differences)

  **Corrected**

- Figure 1: please specify the months too, left and right.

  **Corrected**

- 302 "range 0" –> range from 0

**Corrected**

- Figure 2: trendline? Please remove it.

**Corrected**

- 353: rephrase it please. It is not the "comparison" that reduces the improvement, but is the improvement that is less strong when considering all stations.

**Corrected**

- 370 attribute to→ explained by

**Corrected**

- 374 SEVIRI → SEVERI as shown in Fig …

**Corrected**

- 404: inactivity → activity

**Corrected**

- in 3.3: locality → location

**Corrected**

- 417-423: please review it, it does not read too well

**Corrected**

- 424: instruments → datasets

**Corrected**

- 446: report → paper

**Corrected**

- 449 was more profound → was only?

**Corrected**

- 451: Meanwhile?

**Corrected**

- Fig 2: too low resolution.

**Corrected**

**References used in the reply**

Barnard, J.C., Fast, J.D., Paredes-Miranda, G., Arnott, W.P. and Laskin, A., 2010. Evaluation of the WRF-Chem" Aerosol Chemical to Aerosol Optical Properties" Module using data from the MILAGRO campaign. Atmospheric Chemistry and Physics, 10(15), pp.7325-7340.

Chaibou, A.A.S., Ma, X., Kumar, K.R., Jia, H., Tang, Y. and Sha, T., 2020. Evaluation of dust extinction and vertical profiles simulated by WRF-Chem with CALIPSO and AERONET over North Africa. Journal of Atmospheric and Solar-Terrestrial Physics, 199, p.105213.

Ginoux, P., Chin, M., Tegen, I., Prospero, J. M., Holben, B., Dubovik, O., and Lin, S.: Sources and distributions of dust aerosols simulated with the GOCART model, J. Geophys. Res., 106, 20225–20273, 2001.

Rennie, M.P., Isaksen, L., Weiler, F., de Kloe, J., Kanitz, T. and Reitebuch, O., 2021. The impact of Aeolus wind retrievals on ECMWF global weather forecasts. Quarterly Journal of the Royal Meteorological Society, 147(740), pp.3555-3586.

Zhao, C., Liu, X., Leung, L.R., Johnson, B., McFarlane, S.A., Gustafson Jr, W.I., Fast, J.D. and Easter, R., 2010. The spatial distribution of mineral dust and its shortwave radiative forcing over North Africa: modeling sensitivities to dust emissions and aerosol size treatments. Atmospheric Chemistry and Physics, 10(18), pp.8821-8838.